# A Rab6 to Rab11 transition is required for dense-core granule and exosome biogenesis in *Drosophila* secondary cells

Adam Wells[1], Cláudia C. Mendes[1,a], Felix Castellanos[1], Phoebe Mountain[1], Tia Wright[1], S. Mark Wainwright[1], M. Irina Stefana[1,b], Adrian L. Harris[2], Deborah C. I. Goberdhan[1,c], Clive Wilson[1]*

1 Department of Physiology Anatomy and Genetics, University of Oxford, Oxford, United Kingdom,
2 Department of Oncology, University of Oxford, Oxford, United Kingdom

☉ These authors contributed equally to this work.
¤a Current address: Aligning Science Across Parkinson's (ASAP) Collaborative Research Network, Chevy Chase, Maryland, United States of America
¤b Current address: Wellcome Centre for Human Genetics, University of Oxford, Oxford, United Kingdom
¤c Current address: Nuffield Department of Women's and Reproductive Health, Women's Centre, University of Oxford, Headington, Oxford, United Kingdom
* clive.wilson@dpag.ox.ac.uk

**Data Availability Statement:** All relevant data are within the manuscript and its Supporting Information files.

## Abstract

Secretory cells in glands and the nervous system frequently package and store proteins destined for regulated secretion in dense-core granules (DCGs), which disperse when released from the cell surface. Despite the relevance of this dynamic process to diseases such as diabetes and human neurodegenerative disorders, our mechanistic understanding is relatively limited, because of the lack of good cell models to follow the nanoscale events involved. Here, we employ the prostate-like secondary cells (SCs) of the *Drosophila* male accessory gland to dissect the cell biology and genetics of DCG biogenesis. These cells contain unusually enlarged DCGs, which are assembled in compartments that also form secreted nanovesicles called exosomes. We demonstrate that known conserved regulators of DCG biogenesis, including the small G-protein Arf1 and the coatomer complex AP-1, play key roles in making SC DCGs. Using real-time imaging, we find that the aggregation events driving DCG biogenesis are accompanied by a change in the membrane-associated small Rab GTPases which are major regulators of membrane and protein trafficking in the secretory and endosomal systems. Indeed, a transition from *trans*-Golgi Rab6 to recycling endosomal protein Rab11, which requires conserved DCG regulators like AP-1, is essential for DCG and exosome biogenesis. Our data allow us to develop a model for DCG biogenesis that brings together several previously disparate observations concerning this process and highlights the importance of communication between the secretory and endosomal systems in controlling regulated secretion.

**Funding:** We acknowledge the support of: Biotechnology and Biological Sciences Research Council - BB/K017462/1, BB/N016300/1, BB/W00707X/1 and BB/W015455/1 to CW; BB/R004862/1 to CW and DCIG: Cancer Research UK - C19591/A19076 to DCIG, ALH, CW; C602/A18974 to ALH: Biochemical Society Krebs Memorial Fund studentship to AW: MINCIENCIAS, Colombia (Call 529) studentship to FC. Wellcome - 091911/B/10/Z and 107457/Z/15/Z for funding to Micron imaging facility. The funders had no role in study design, data collection and analysis, decision to publish, or preparation of the manuscript.

**Competing interests:** The authors have declared that no competing interests exist.

## Author summary

Cells communicate with each other by releasing signalling molecules that bind receptors on target cells and alter their behaviour. Before their release, these signals are typically stored in condensed structures called dense-core granules (DCGs). DCGs are found in many animal species and their dysregulation is linked to several major diseases, such as diabetes and neurodegenerative disorders. However, the mechanisms controlling DCG formation and secretion are only partly understood. Here we study this process in fruit flies using a secretory cell that contains unusually large DCGs. We show that known regulators of DCG formation in mammals also control DCG production in these fly cells and identify new assembly steps by following the process in living cells. Most importantly, we show that an interaction between the cell's secretory compartments and its recycling endosomal compartments is required to induce within minutes the condensation of proteins into a DCG. We further find that known regulators of DCG formation are needed for this crucial interaction to take place. Our work provides a platform, which future studies can build upon, to uncover the molecular mechanisms that enable this critical secretory-endosomal interaction and probe its role in diseases of secretion.

## Introduction

Proteins are secreted from eukaryotic cells by both constitutive and regulated pathways. Hormone-, enzyme- and neuropeptide-producing cells, including endocrine adrenal chromaffin cells and pancreatic β cells, exocrine pancreatic acinar cells and neurons, respectively, are typically specialised for regulated secretion. They package and store the proteins, which they will release, in dense-core granules (DCGs) within so-called dense-core vesicles [1].

By studying the DCG packaging process in cell lines that represent some of these different secretory cell types (e.g. [2–4]), several mechanisms controlling DCG biogenesis and release have been identified and shown to be shared between different cells. For example, clustering of cargos into immature DCG compartments in the *trans*-Golgi network requires cholesterol and lipid raft-like structures [3], together with specific enzymes that regulate lipid metabolism, namely phospholipase D1 and diacylglycerol kinase [5]. Granin proteins may be required for compartment budding and DCG assembly [6,1]. Several cytosolic adaptor proteins, such as the AP-1 coatamer complex and Golgi-localising, γ-adaptin ear homology domain, ARF-binding proteins (GGAs) are recruited to the *trans*-Golgi by the small G protein Arf1 [7,8]. They are then involved in trafficking molecules to and from maturing DCG compartments [9]. The maturation process is also dependent on reduced pH [10] and an increase in intraluminal calcium ($Ca^{2+}$) ions [11]. However, the molecular and membrane trafficking processes that drive and coordinate these changes remain unclear. Once matured, the DCG compartments fuse to the plasma membrane via mechanisms requiring $Ca^{2+}$-dependent synaptotagmins and vesicle-associated membrane proteins (VAMPs) [12].

Rab GTPases are another key set of molecules involved in regulated secretion. These small monomeric GTPases control membrane trafficking and organelle identity in the secretory and endolysosomal systems [13]. Analysis in mammalian cell lines has highlighted a role for Rab6 in the formation of immature DCG compartments at the *trans*-Golgi network [14]. However, mature dense-core vesicles have been reported to be associated with endosomal Rabs, like Rab11 [15], suggesting a potential cross-talk between the secretory and endosomal systems in regulated secretion.

Dissecting out the molecular mechanisms underlying regulated secretion is not only of biological importance, but of clinical relevance. For example, Type 1 and Type 2 diabetes involves aberrant secretory regulation in β cells [16,17]. Defective processing and secretion of specific proteins, such as the Amyloid Precursor Protein (APP), is associated with neuro-degenerative diseases, including Alzheimer's Disease [18,19]. In mammals, however, cell biological studies of regulated secretion are usually undertaken in cultured cells, either isolated primary cells or immortalised cell lines, where it is difficult to reproduce the physiological microenvironment in which secretory cells function in living organisms. Furthermore, it is challenging to follow the events involved, because high-resolution imaging of DCG compartments, each of which is typically 0.2–1 μm in diameter, traditionally requires electron microscopy on fixed tissue.

The larval salivary gland [20–23]and secretory cells of the proventriculus [24] in the fruit fly, *Drosophila melanogaster*, as well as neurons and neuromuscular junctions in flies [25] and the nematode, *Caenorhabditis elegans* [26], have provided *in vivo* genetic systems to study DCG formation and release in invertebrates, with several novel conserved regulators identified in *C. elegans* (e.g. [27]. Generally, the DCG compartments in these cells are small and their substructures difficult to resolve with light microscopy. However, salivary gland granules increase in size during maturation, and it has therefore been possible to follow and genetically dissect some steps in this process [28]. Such studies have uncovered, for example, that Rab1 and Rab6 localise to the membranes of maturing secretory granules, while Rab11 and Rab1 drive granule growth and maturation via mechanisms that are yet to be characterised [29,30].

We have characterised another cell system in *Drosophila*, the secondary cell (SC) of the male accessory gland, as a genetic cell model for regulated secretion (Fig 1A, 1B and 1B'; [31–34]. Its secretory compartments are exceptionally large, approximately 5 μm in diameter, making it possible to visualise the structures formed within. These include a ~3 μm diameter DCG [32], surrounded by intraluminal vesicles (ILVs), which are released as exosomes upon compartment fusion with the plasma membrane [31,33]. ILV formation in SCs has been found to require the core endosomal complexes required for transport (ESCRT) proteins as well as specific accessory ESCRT proteins, some of which also affect DCG biogenesis [34]. Some immature secretory compartments, which lack DCGs, are marked by Rab6, while a few compartments that are less well characterised are coated by Rab19. However, all mature DCG compartments carry Rab11 at their surface [35,33], further supporting a link between endosomal and secretory pathways in DCG biogenesis. Although only a small subset of ILVs formed inside these latter compartments are labelled by Rab11, the entire population of exosomes that these compartments produce is referred to as Rab11-exosomes [33]. The Rab11-exosome pathway and its regulation appear to be conserved from fly to human cells, and Rab11-exosomes have important physiological and pathological functions [34], making the further characterisation of this trafficking route of considerable interest.

Here, we employ genetic knockdown techniques in SCs to reveal parallels between *Drosophila* SCs and mammalian cells in the regulation of DCG biogenesis. Furthermore, by visualising the process of DCG formation in real-time, we show that it is accompanied by a Rab6 to Rab11 transition, and not by trafficking of cargos from Rab6-positive to Rab11-positive compartments. This Rab transition, which is controlled by known regulators of mammalian DCG biogenesis, plays a critical role in Rab11-exosome formation, but also DCG biogenesis, presumably by establishing a DCG-inducing microenvironment inside these compartments.

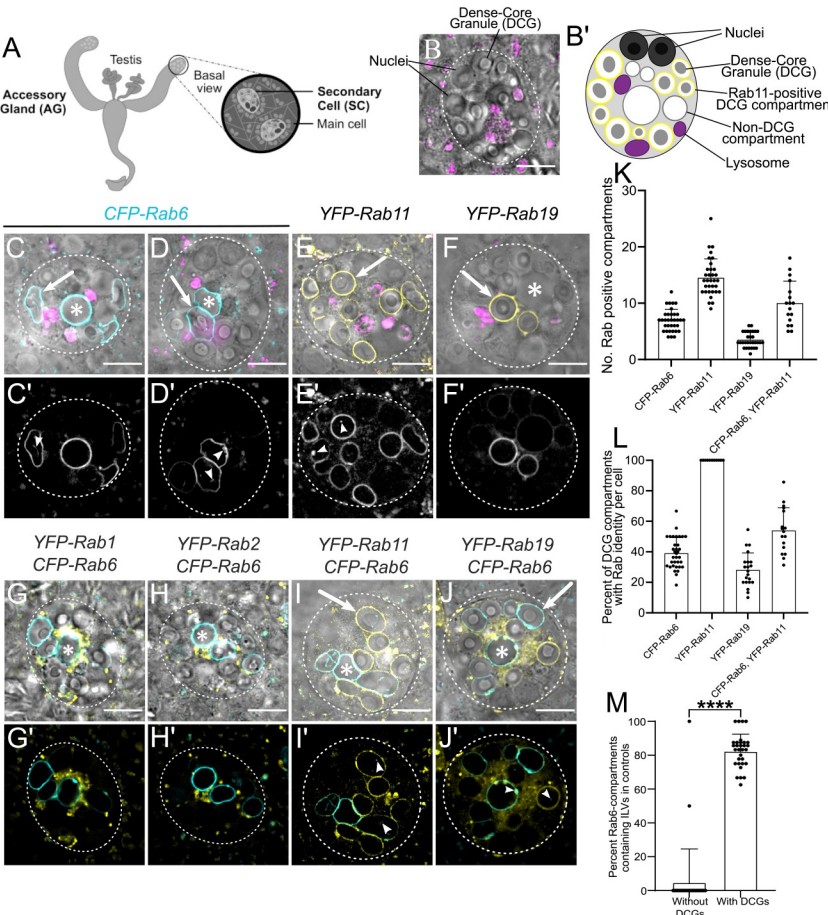

**Fig 1. Morphology and Rab identity of DCG compartments in *Drosophila* secondary cells.** (A) Schematic of *Drosophila* accessory gland (AG), showing secondary cells (SCs) at the distal tip of each lobe. (B) *Ex vivo* Differential Interference Contrast (DIC) image of an SC from the AG of a six-day-old $w^{1118}$ virgin male fly, stained with LysoTracker Red. (B') Schematic of SC, with equivalent structures labelled. (C-J) DIC images of SCs, overlaid with LysoTracker Red (C-F), and fluorescent signal from different endogenously tagged *Rab* genes (C-J). DIC images: grayscale, YFP: yellow, CFP: cyan, LysoTracker Red: magenta. (C'-J') Fluorescence-only images showing expression of each Rab. Arrowheads indicate ILVs labelled by various Rabs, which lie inside compartments. (C, D) CFP-Rab6-labelled compartments have a range of morphologies including spherical non-DCG compartments (C, *), irregularly shaped, non-DCG compartments (D, *), and DCG-containing compartments (arrows). (E) YFP-Rab11 marks all DCG compartments. (F) YFP-Rab19 marks two or three DCG compartments. (G, H and G', H') Small YFP-Rab1- (G) and YFP-Rab2-positive (H) clustered compartments surround central non-DCG-containing, CFP-Rab6 compartments. (I) Some YFP-Rab11-positive DCG compartments are labelled with CFP-Rab6. CFP-Rab6 puncta are also observed inside some compartments that are not Rab6-positive (arrow) and in a compartment that is Rab6- and Rab11-positive, but lacks a DCG (I, *). (J) YFP-Rab19 compartments are not co-labelled with CFP-Rab6. However, YFP-Rab19 and CFP-Rab6 do co-label microdomains and internal membranes on Rab6-marked compartments, eg. arrowhead in centre (J'), while Rab6-positive internal puncta are also found inside Rab19-positive compartments (arrowhead on right). (K) Bar chart showing number of large compartments (> 1 μm diameter at its widest point) positive for different fluorescent Rabs in individual SCs. (L) Bar chart showing % of DCG compartments labelled with specific fluorescent Rabs in individual SCs. (M) Bar chart showing the proportion of CFP-Rab6-positive DCG compartments and coreless CFP-Rab6 compartments containing Rab6-positive ILVs in individual SCs expressing a control *rosy*-RNAi. Data for bar charts were collected from three SCs per gland derived from 10 glands, except for the genotype expressing CFP-Rab6 and YFP-Rab11, where the relative expression levels of both fusion proteins varied considerably between different cells, so only some cells were suitable for analysis. Approximate outlines of SCs are marked by dashed circles. Scale bars: 10 μm. * marks representative non-acidic compartments that lack a DCG. Arrows mark representative DCG compartments labelled by various Rabs. For K-M, bars show mean ± SD; CFP-Rab6, n = 34; YFP-Rab11, n = 33; YFP-Rab19, n = 30; CFP-Rab6/YFP-Rab11, n = 17; for M, n = 31; P<0.0001: ****. Genotypes for images: (C-D) $w^{1118}$; $TI\{TI\}Rab6^{EYFP}/+$; (E) $w^{1118}$; $TI\{TI\}Rab11^{EYFP}/+$; (F) $w^{1118}$; $TI\{TI\}Rab19^{EYFP}/+$; (G) $w^{1118}$; $TI\{TI\}Rab6^{EYFP}/+$; $w^{1118}$; $TI\{TI\}Rab1^{EYFP}/+$; (H) $w^{1118}$; $TI\{TI\}Rab6^{EYFP}/TI\{TI\}Rab2^{EYFP}$; (I) $w^{1118}$; $TI\{TI\}Rab6^{EYFP}/+$; $TI\{TI\}Rab11^{EYFP}/+$; and (J) $w^{1118}$; $TI\{TI\}Rab6^{EYFP}/+$; $TI\{TI\}Rab19^{EYFP}/+$.

## Results

### Rab6, Rab11 and Rab19 mark specific compartment subsets in the SC secretory pathway

A previous expression analysis, primarily in fixed SCs, using endogenously tagged *Rab* genes, where a YFP-Rab fusion protein is produced at normal levels from the endogenous *Rab* locus [35], confirmed that DCG compartments are marked by Rab11 [32]. However, it also revealed that some secretory compartments are labelled by Rab6 and by Rab19, a poorly characterised Rab proposed to be involved in apical secretion [36]. To assess the overlaps in expression of these different Rabs in more detail, we employed a live imaging approach, where the morphology of SCs is much better preserved. The fluorescently tagged Rabs, which were either CFP- or YFP-Rab fusion proteins, had no effect on the number of DCG compartments in SCs, as scored using differential interference contrast (DIC) microscopy (Figs 1B and S1A).

In these and subsequent experiments, we analysed SC morphology in males aged at 29˚C following eclosion, because this allowed us to use a temperature-controlled gene expression system to knock down gene function in SCs specifically in adults. Previous studies have indicated that this does not affect SC morphology and males remain fertile (eg. [31,33,34]). SCs contained 11 ± 2.2 (mean ± SD) DCG compartments (n = 75), none of which were stained by the acidic dye LysoTracker Red (eg. Figs 1C–1F; S1B and S1C), and an additional 2.7 ± 2.0 (n = 34) large non-acidic compartments that did not contain a DCG. A CFP-tagged form of Rab6 localised to 7.2 ± 1.9 (n = 34) non-acidic compartments, including all those that lacked a DCG (Fig 1C, 1D and 1K). Some of these latter compartments were enlarged and/or irregular in shape, though frequently, at least one spherical non-DCG compartment of similar size to the DCG compartments was positioned centrally within the SC (Fig 1C). We confirmed that the numbers of large Rab-labelled non-acidic and DCG compartments observed in adults shifted to 29˚C at eclosion was unchanged when compared to adult males maintained at 19˚C (S1D–S1G Fig).

Rab1 and Rab2 are two Golgi-associated Rabs that have been implicated in trafficking processes that take place around the *trans*-Golgi network and precede formation of DCGs. In yeast, Rab1-labelled Golgi membranes convert to Rab6-coated membranes at the *trans*-Golgi [37], while in nematodes, Rab2-interacting proteins associated with the Golgi apparatus are required for DCG compartment formation [26,27]. Both YFP-Rab1 (Fig 1G) and YFP-Rab2 (Fig 1H) fusion proteins, when expressed from their endogenous gene loci, were concentrated in Golgi-like cisternae around the Rab6-positive, non-DCG-containing central compartments of SCs, consistent with these latter compartments being generated at the surface of the *trans*-Golgi. Indeed, staining with an anti-GM130 antibody, which primarily marks *cis*-Golgi compartments, revealed some co-localisation with both Rab1 and Rab2 (S2 Fig). However, there were also adjacent GM130-negative Rab1- and Rab2-positive compartments, particularly for Rab1, in both SCs and adjacent main cells, which presumably represent *trans*- and medial-Golgi cisternae.

As previously reported [33], YFP-Rab11 marked all the large DCG-containing compartments in SCs (Fig 1E and 1L). About 39.0 ± 10.1% of these were also marked by Rab6 (Fig 1I, 1K and 1L). Occasionally, one large non-acidic compartment that lacked a DCG was also weakly labelled with Rab11 and it often contained internal CFP-Rab6 puncta (Fig 1I). Based on our previous findings using endogenously tagged *YFP-Rab11* as a marker [33], we conclude that a small fraction of the Rab6 found at the compartmental limiting membrane is presumably packaged inside ILVs. CFP-Rab6 puncta were also observed in some DCG compartments that lacked CFP-Rab6 at their surface (Figs 1I and S1H).

ILVs marked by CFP-Rab6 and/or YFP-Rab11 were often seen adjacent to the boundaries of DCGs (S1B Fig) and in some cases, extended along an arc around the DCG surface

(S1C Fig). Rab-labelled ILVs can also form continuous bridge-like structures running from the compartment's limiting membrane to the DCG, sometimes across a gap of one to two micrometres (S1B and S1C Fig). This observation was previously reported, when using membrane-associated exosome markers in SCs with more ubiquitous ILV-labelling profiles [33,38]. Although the significance of this DCG association remains unclear, it is notable that ILV puncta appeared almost exclusively in compartments with DCGs. In SCs, over 80% of CFP-Rab6-labelled compartments with DCGs also contained Rab6-positive ILVs (Fig 1M). By contrast, only a few core-less Rab6-labelled compartments contained ILVs, with the vast majority of cells containing only ILV-deficient, core-less compartments.

Finally, YFP-Rab19 labelled the entire surface of 3.7 ± 1.3 large compartments (n = 30), all of which contained DCGs (Fig 1F, 1K and 1L). These compartments were never marked by CFP-Rab6 (Fig 1J), although YFP-Rab19 did mark microdomains on the surface of other non-acidic compartments, including those labelled by CFP-Rab6 (Figs 1J and S1I). In addition to these microdomains on compartment surfaces, YFP-Rab19 also localised internally within various compartments, including those not marked by Rab19 on their surface (Fig 1J'). This indicates that Rab19 can be incorporated into specific populations of ILVs, perhaps through invagination of the microdomains we observe on the surface of some compartments.

In summary, Rab6, Rab11 and Rab19 mark the highly enlarged secretory compartments found in SCs. Rab6 labels all large non-acidic compartments that do not contain DCGs, whilst Rab11 marks all the compartments that do. There is also significant overlap between these two markers, with Rab6 and Rab11 often colocalising on DCG compartments and on sporadic non-DCG compartments. Rab19 marks a subset of DCG compartments that is distinct from the subset of Rab6-labelled compartments.

## Small Rab1-positive compartments give rise to Rab6-positive endosomes

To begin to understand the dynamics of secretory compartment biogenesis in SCs, it was first important to determine how the various compartments we had identified related to each other. We therefore began by investigating the relationship between the small Rab1-positive compartments that contact larger Rab6-positive compartments (Fig 1G). Since Rab1 is frequently associated with Golgi stacks, whilst Rab6 is often located and functions in the *trans*-Golgi network [14], we hypothesised that the Rab1- and Rab6-labelled compartments in SCs might interact, either through trafficking between these compartment types or by Rab1-positive compartments maturing into Rab6-positive compartments, as observed in yeast [37]. To test this, we carried out time-lapse imaging of live SCs expressing *YFP-Rab1* and *CFP-Rab6* from the endogenous *Rab* loci.

Through these experiments, it became clear that Rab1-positive compartments directly transition into Rab6-positive ones as they mature (Fig 2, S1 Movie). During this process, a small Rab1-labelled compartment increases in size from approximately 0.5 µm in diameter (Fig 2A and 2B), until it has reached between 2.5 µm and 10 µm in diameter. Concurrently with this dramatic increase in size, the strong labelling by Rab1 is gradually lost and the compartment accumulates Rab6 on its limiting membrane to form a large Rab6-positive compartment that lacks a DCG (Fig 2D). This process takes between 30 and 60 minutes, and soon after its completion, the Rab6-positive compartment produced contracts in volume again, after which it can form a DCG (Fig 2E). Therefore, time-lapse experiments not only demonstrate that Rab1-positive compartments are the direct precursors of Rab6-positive compartments, but also that they are the same compartments that will eventually go on to form DCGs.

We also investigated what mechanisms might permit small Rab1-positive compartments to grow so rapidly. Through further time-lapse imaging, we found that jointly labelled Rab1/

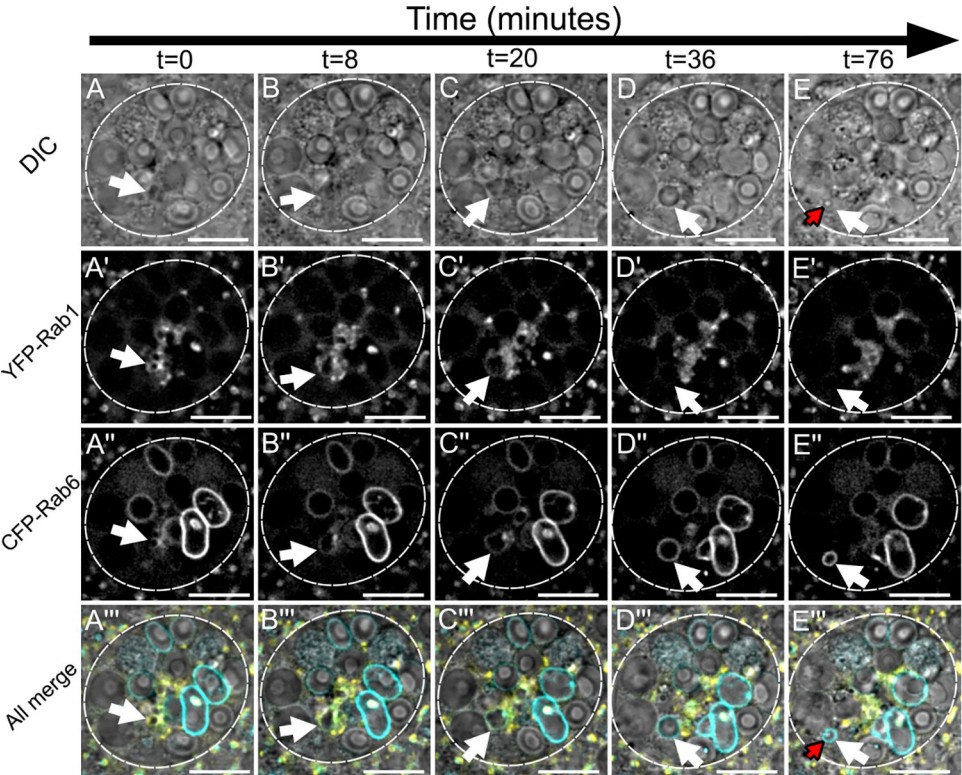

**Fig 2. A Rab1 to Rab6 transition accompanies the maturation of secretory compartments at the *trans*-Golgi network of *Drosophila* SCs (related to S1 Movie).** Panels show *ex vivo* images of a single SC taken at five discrete timepoints with time since start of imaging shown above in minutes. Rows within panel display cellular organisation at each timepoint through DIC imaging (A-E), fluorescent YFP-Rab1 signal (A'-E'), fluorescent CFP-Rab6 signal (A"-E"), and combined images displaying all three (A'''-E'''). White arrows indicate the position of a secretory compartment as it matures through a Rab1 to Rab6 transition across time and red arrows indicate the position of a newly formed DCG inside that compartment. (A-C) A small, central compartment (white arrow; <1μm diameter) that is primarily labelled with YFP-Rab1 grows rapidly in size, losing most of the YFP-Rab1 signal from its surface and accumulating more CFP-Rab6. (D) The compartment loses all detectable YFP-Rab1 signal from its surface, obtains its greatest diameter, becomes perfectly spherical and starts to migrate peripherally. (E) The compartment retains its CFP-Rab6 identity but begins to contract again in diameter, as a DCG rapidly appears inside it (red arrow). The time interval between the formation of a large Rab6-positive compartment and DCG biogenesis varies between compartments, with this example being particularly rapid. Approximate outline of SC is marked by dashed circles. Scale bars: 10 μm. This Rab transition was observed four times with different accessory glands. Genotype for images: $w^{1118}$; TI{TI}Rab6$^{CFP}$/+; TI{TI}Rab1$^{EYFP}$/+.

Rab6-compartments are able to fuse together, thereby increasing their size as they mature (S3 Fig, S2 Movie). Since later compartments marked solely by Rab6 do not appear to be capable of homotypic fusion, our findings suggest that either Rab1 or a combination of Rab1 and Rab6 are required for these fusion events to occur.

## A Rab6 to Rab11 transition accompanies DCG formation in SCs

We have previously shown that a new DCG compartment is generated every 4–6 h in SCs [32]. Having demonstrated that Rab11 marks all DCG compartments and that Rab6 and Rab11 colocalise on some DCG compartments and occasionally on one large non-acidic, core-less compartment, we investigated whether Rab6-positive compartments that lack DCGs might be the precursors of Rab11-marked DCG compartments, using flies expressing both the *CFP-Rab6* and *YFP-Rab11* fusion genes.

These experiments showed a clear transition from Rab6- to Rab11-labelling on secretory compartments, a process that occurred over the course of many hours (Figs 3 and S4, S3 and S4 Movies). This transition took place in several stages and coincided with a number of important changes in the compartment. At the earliest stage, compartments marked by Rab6 lack a DCG and have no Rab11 present on their limiting membranes (Fig 3A, white arrow, and S4A Fig). Subsequently, Rab11 begins to be recruited and this coincides with the compartment contracting in size, the beginning of ILV biogenesis, and the appearance of transient dense accumulations of Rab6 on the inside of the limiting membrane (Figs 3B and S4B). It also immediately precedes the formation of DCGs inside compartments (Fig 3C). As well as occurring soon after the recruitment of Rab11, DCG biogenesis is completed rapidly, typically in less than 20 min (S3 Movie). Following DCG biogenesis, maturing secretory compartments continue to recruit Rab11 to their membranes whilst gradually losing Rab6, resulting in DCG compartments that are labelled primarily by Rab11, but may still contain Rab6-positive ILV puncta (Figs 3D–3F and S4C). These persist, even after Rab6 has been removed from the compartment's limiting membrane (Figs 1I and S1H).

## DCG biogenesis in SCs is regulated by evolutionarily conserved mechanisms

The changes in Rab identity observed when DCG compartments mature in SCs are consistent with some of the previously reported Rab identities of secretory compartments at different maturation stages in secretory cells of flies and other organisms [26,27,37,29]. Despite the remarkably large size of SC DCGs, we reasoned that DCG compartments in these cells were likely to be assembled via similar mechanisms to those employed in mammalian cells. To test this, we assessed the effects of knocking down two conserved trafficking regulators involved in mammalian DCG biogenesis, *Arf1* (otherwise called *Arf79F*) and components of the AP-1 coatomer complex. Both have established roles in regulating maturation and appropriate cargo loading in DCG compartments (e.g. [7–9]). To mark dense cores, we expressed a GPI-anchored form of GFP, which concentrates in the DCGs of SCs as they mature (Fig 4A; [32]), using the GAL4/UAS modular gene expression system [39].

To focus our analysis on the maturation of DCG compartments, we suppressed gene expression in SCs exclusively in adults, when these compartments start to form, but not during SC development (see also [31,33,34]). To do this, we expressed RNAis under the control of *dsx*-GAL4, a driver line expressed only in SCs within the male accessory gland. The flies also expressed a temperature-sensitive form of the GAL4 inhibitor, GAL80ts, in all cells. They were maintained at 25˚C until eclosion of the adult, then shifted to 29˚C, which is the non-permissive temperature for GAL80, thus activating RNAi expression. A disadvantage of this cell type-specific knockdown is that SCs make up less than 4% of all cells in the accessory gland, so we could not confirm the levels of knockdown by qRT-PCR or western analysis. However, all RNAis, other than the negative control, which targeted *rosy* (*ry*), a gene involved in eye pigmentation that encodes xanthine dehydrogenase, induced phenotypes.

To control for off-target effects, we employed at least two independent RNAis or RNAis for three different subunits of the AP-1 complex. All of these RNAis have been used in previous studies (see Materials and Methods). For each gene or protein complex, at least one RNAi had previously been shown to produce strong and specific phenotypes, and in our hands, these RNAis typically induced the strongest phenotypes. However, the other RNAis produced related, though often weaker, phenotypes, indicating that the effects observed were target gene-specific.

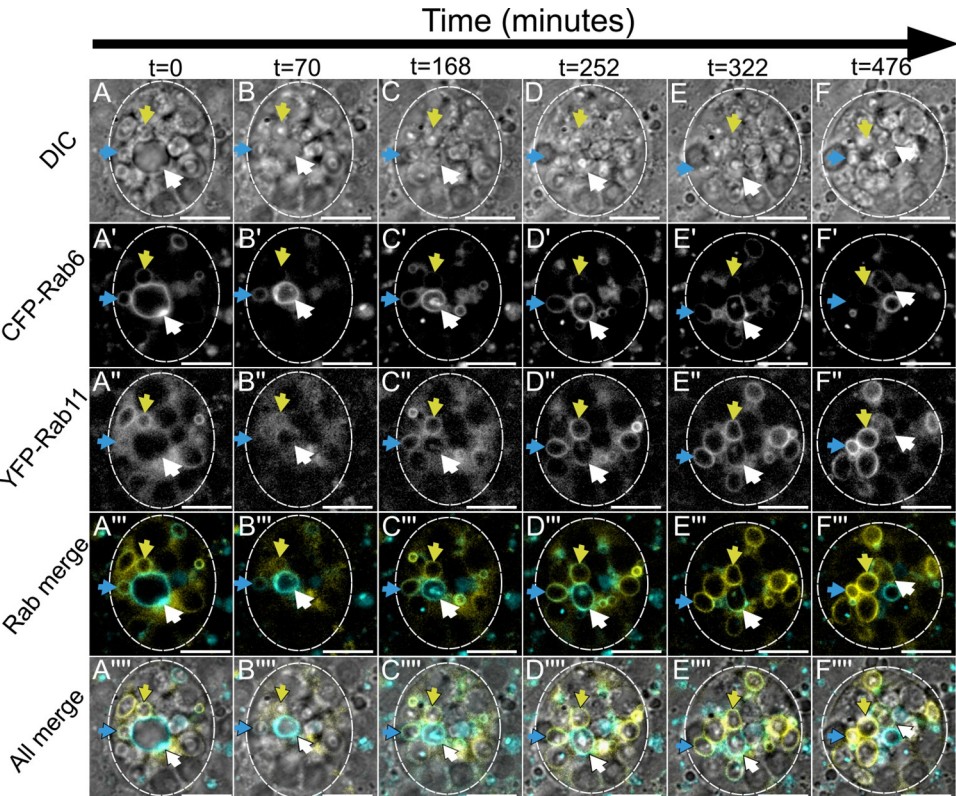

**Fig 3. Rab6 to Rab11 transition on surface of maturing secretory compartments in *Drosophila* secondary cells coincides with exosome and DCG biogenesis (related to S3 Movie).** Panel shows *ex vivo* images of a single SC taken at six discrete timepoints with time since start of imaging shown above in minutes. Rows within panel display cellular organisation at each timepoint through DIC imaging (A-F), fluorescent CFP-Rab6 signal (A'-F'), fluorescent YFP-Rab11 signal (A"-F"), a combined fluorescence image (A'''-F''') and a combined DIC and fluorescence image (A''''-F''''). Three coloured arrows (blue, yellow and white) each indicate the position of one maturing secretory compartment across time. (A-A''') The compartments marked by either a blue or yellow arrow begin with CFP-Rab6 and YFP-Rab11 co-labelling and have DCGs already present. The compartment marked by the white arrow is significantly larger, is labelled strongly with CFP-Rab6, but has no YFP-Rab11 on its surface and no DCG. (B-B''', C-C''') The blue and yellow arrowed compartments lose CFP-Rab6 labelling over time and become more heavily marked by YFP-Rab11. The compartment marked with a white arrow significantly contracts in size and is only weakly labelled by CFP-Rab6 by the end of the time course. In contrast, YFP-Rab11 begins to accumulate on the compartment. Simultaneously, this compartment begins forming internal structures, with ILVs appearing first (B') and then a DCG (C). ILVs are marked by both CFP-Rab6 and YFP-Rab11 and at least partly surround the DCG (C). (D-D''', E-E''', F-F''''). The two more mature highlighted compartments (marked with blue and yellow arrows) lose CFP-Rab6 identity and are strongly labelled by YFP-Rab11 by the end of the time course. The compartment labelled with a white arrow still retains some CFP-Rab6, but YFP-Rab11 continues to increase in levels. Approximate outline of SC is marked by dashed circles. Scale bars: 10 μm. This Rab transition and concurrent DCG formation was observed four times with different accessory glands. Genotype for images: $w^{1118}$; $TI\{TI\}Rab6^{CFP}/+$; $TI\{TI\}Rab11^{EYFP}/+$.

Knockdown of *Arf1* with two independent RNAis expressed under the control of the *dsx*-GAL4 driver almost invariably produced SCs without any DCGs, as judged by both GFP-GPI fluorescence and DIC microscopy (Figs 4B, 4F and S5B). *Arf1*-RNAi #1 [40] produced the most penetrant phenotype. In addition, significantly fewer non-acidic compartments were produced, which were often larger than normal and which contained diffuse GFP rather than the concentrated cores of GFP fluorescence seen in normal DCGs (Fig 4B, 4G and 4H). Compared to controls, a greater proportion of large acidic late endosomal and lysosomal compartments also contained GFP (S6A Fig), presumably reflecting more trafficking of GFP-GPI to these compartments and/or reduced GFP quenching or degradation within them.

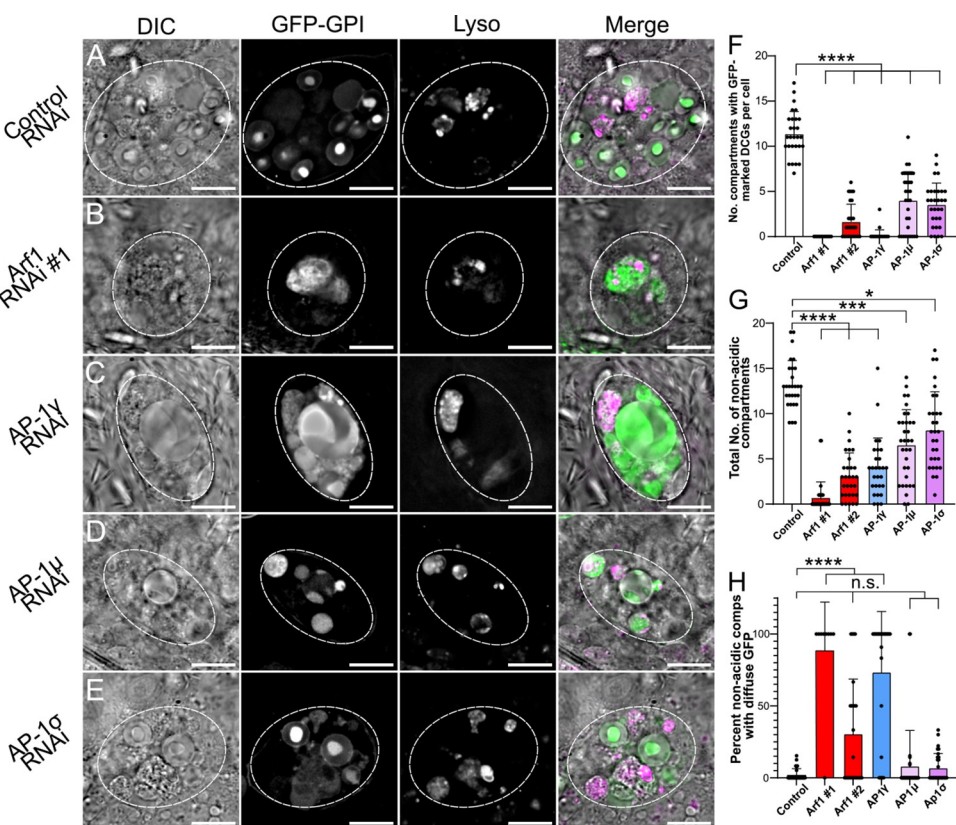

**Fig 4. The conserved trafficking regulators Arf1 and AP-1 are essential for DCG biogenesis in SCs.** (A-E) Representative images of SCs expressing the DCG marker *GFP-GPI* together with a control RNAi (A) or RNAis targeting *Arf1* (RNAi #1; B), *AP-1γ* (C), *AP-1μ* (D) or *AP-1σ* (E). Cellular organisation was assessed using DIC imaging, GFP-GPI fluorescence and Lysotracker Red fluorescence; a merged image is also shown for each cell. Note that the knockdowns generally reduce the number of large non-acidic compartments and the number of DCG compartments, though some remaining compartments can be expanded in size. (F) Bar chart showing number of compartments containing GFP-labelled DCGs in control SCs and following knockdown of *Arf1* and *AP-1* subunits. (G) Bar chart showing number of non-acidic compartments in these different genotypes. (H) Bar chart showing the percentage of non-acidic compartments with diffuse GFP-GPI in these different genotypes. Approximate outlines of SCs are marked by dashed circles. Scale bars: 10 μm. Data are typically for three cells per accessory gland; accessory glands from ≥10 individual males were imaged during three to six separate imaging sessions, for this and subsequent knockdown experiments. For F-H, bars show mean ± SD; Control, n = 28; *Arf1* #1, n = 31; *Arf1* #2, n = 30; *AP-1γ*, n = 28; *AP-1μ*, n = 32; *AP-1σ*, n = 30. P<0.05: *, P<0.01: **, P<0.001: ***, P<0.0001: ****. Genotypes for images: (A) $w^{1118}$; $P\{tub\text{-}GAL80^{ts}\}/P\{ry^{TRiP.HMS02827}\}$; dsx-GAL4, P{UAS-GFP.GPI}/+; (B) $w^{1118}$; $P\{tub\text{-}GAL80^{ts}\}/+$; dsx-GAL4, P {UAS-GFP.GPI}/P{Arf1$^{GD12522}$}; (C) $w^{1118}$; $P\{tub\text{-}GAL80^{ts}\}/+$; dsx-GAL4, P{UAS-GFP.GPI}/P{AP-1γ$^{TRiP.JF02684}$}; (D) $w^{1118}$; $P\{tub\text{-}GAL80^{ts}\}/+$; dsx-GAL4, P{UAS-GFP.GPI}/P{AP-1μ$^{GD14206}$}; (E) $w^{1118}$; $P\{tub\text{-}GAL80^{ts}\}/P\{AP\text{-}1σKK108869\}$; dsx-GAL4, P{UAS-GFP.GPI}/+.

Knockdown of three different subunits of the AP-1 coatomer complex, *AP-1γ*, *AP-1μ* and *AP-1σ*, also strongly and significantly reduced the number of DCG compartments in SCs and the total number of non-acidic compartments (Fig 4C–4G). However, the range of phenotypes was broader for these different RNAis, suggesting either different levels of knockdown or possibly an off-target effect for *AP-1γ* knockdown, which produced the strongest phenotypes. An off-target effect seems unlikely, however, since this *AP-1γ*-RNAi has been used in a previous study where it produced the predicted selective effect on secretory trafficking [41]. Furthermore, following *AP-1γ* knockdown, the non-acidic compartments remaining in SCs frequently contained diffuse GFP, mirroring the phenotype seen with *Arf1* knockdown (Fig 4H). In addition, many phenotypes broadly paralleled those seen with other AP-1 subunit RNAis (see also

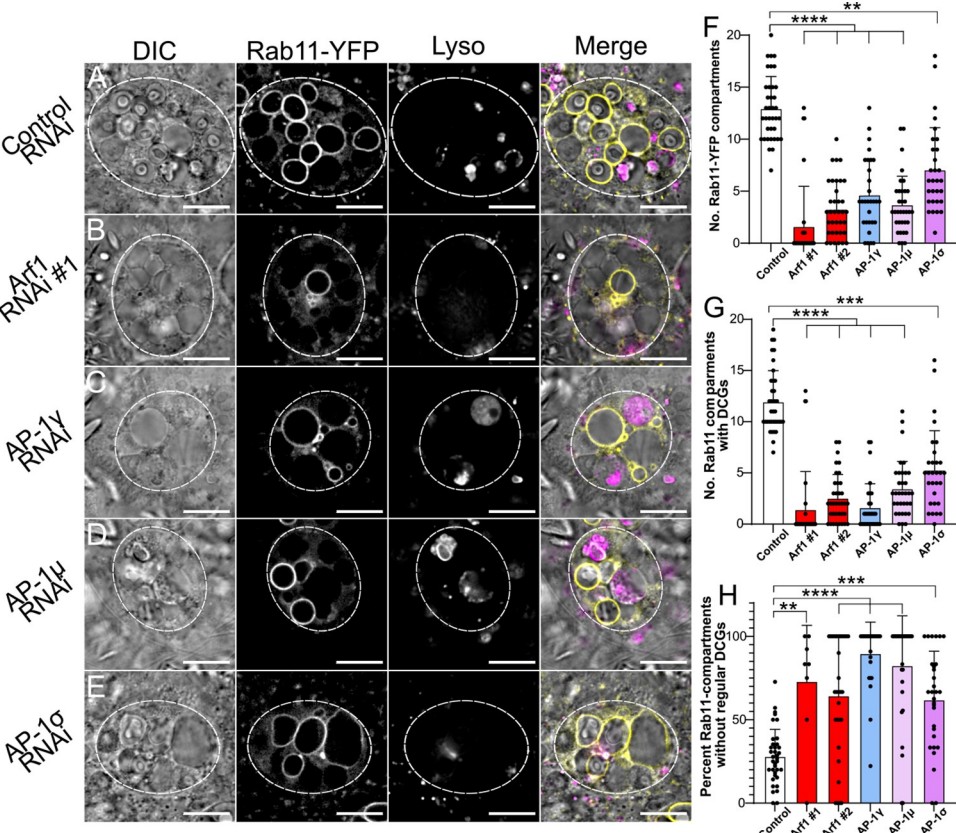

**Fig 5. *Arf1* and *AP-1* regulate Rab11-compartment identity and subsequent DCG biogenesis.** (A-E) Representative images of SCs expressing the *YFP-Rab11* fusion gene from the endogenous *Rab* locus together with a control RNAi (A) or RNAis targeting *Arf1* (RNAi #1; B), *AP-1γ* (C), *AP-1μ* (D) or *AP-1σ* (E). Cellular organisation is assessed using DIC imaging, YFP-Rab11 fluorescence and Lysotracker Red fluorescence, and a merged image is provided for each cell. Note that in the knockdown cells, there are fewer Rab11-positive compartments and more of them either do not contain DCGs, or contain abnormally shaped or multiple DCGs, when compared to controls. (F) Bar chart showing number of YFP-Rab11-labelled compartments in control SCs and following knockdown of *Arf1* and *AP-1*. (G) Bar chart showing the number of YFP-Rab11 compartments containing DCGs in these different genotypes. (H) Bar chart showing the percentage of YFP-Rab11 compartments which fail to produce regularly shaped DCGs in these different genotypes. Approximate outlines of SCs are marked by dashed circles. Scale bars: 10 μm. For F-H, bars show mean ± SD; Control, n = 35; *Arf1* #1, n = 31; *Arf1* #2, n = 36; *AP-1γ*, n = 30; *AP-1μ*, n = 34; *AP-1σ*, n = 30. P<0.05: *, P<0.01: **, P<0.001: ***, P<0.0001: ****. Genotypes for images: (A) *w1118; P{tub-GAL80ts}/P{ryTRiP.HMS02827}; dsx-GAL4, TI{TI}Rab11EYFP/+*; (B) *w1118; P{tub-GAL80ts}/+; dsx-GAL4, TI{TI}Rab11EYFP/P{Arf1GD12522}*; (C) *w1118; P{tub-GAL80ts}/+; dsx-GAL4, TI{TI}Rab11EYFP/P{AP-1γTRiP.JF02684}*; (D) *w1118; P{tub-GAL80ts}/+; dsx-GAL4, TI{TI}Rab11EYFP/P{AP-1μGD14206}*; (E) *w1118; P{tub-GAL80ts}/P{AP-1σKK108869}; dsx-GAL4, TI{TI}Rab11EYFP/+*.

Figs 5 and 6). For example, a high proportion of acidic compartments contained fluorescent GFP for all three *AP-1* knockdowns, as also observed with *Arf1* knockdowns (S6A Fig).

In mammalian cells, Arf1 and the AP-1 complex are thought to control the formation of DCG compartments and subsequent cargo loading respectively. In view of our findings that this process involves a Rab6 to Rab11 transition in SCs, we tested whether Arf1 and AP-1 control this transition. For this, we knocked down *Arf1* and *AP-1* components in SCs expressing either the endogenous *CFP-Rab6* or *YFP-Rab11* fusions. Interestingly, although the fly lines employed drive RNAi expression under the control of *dsx*-GAL4 in the same way as the GFP-GPI line we had used, the level of knockdown appeared to be reduced and some DCG compartments were generated in many of these genetic backgrounds.

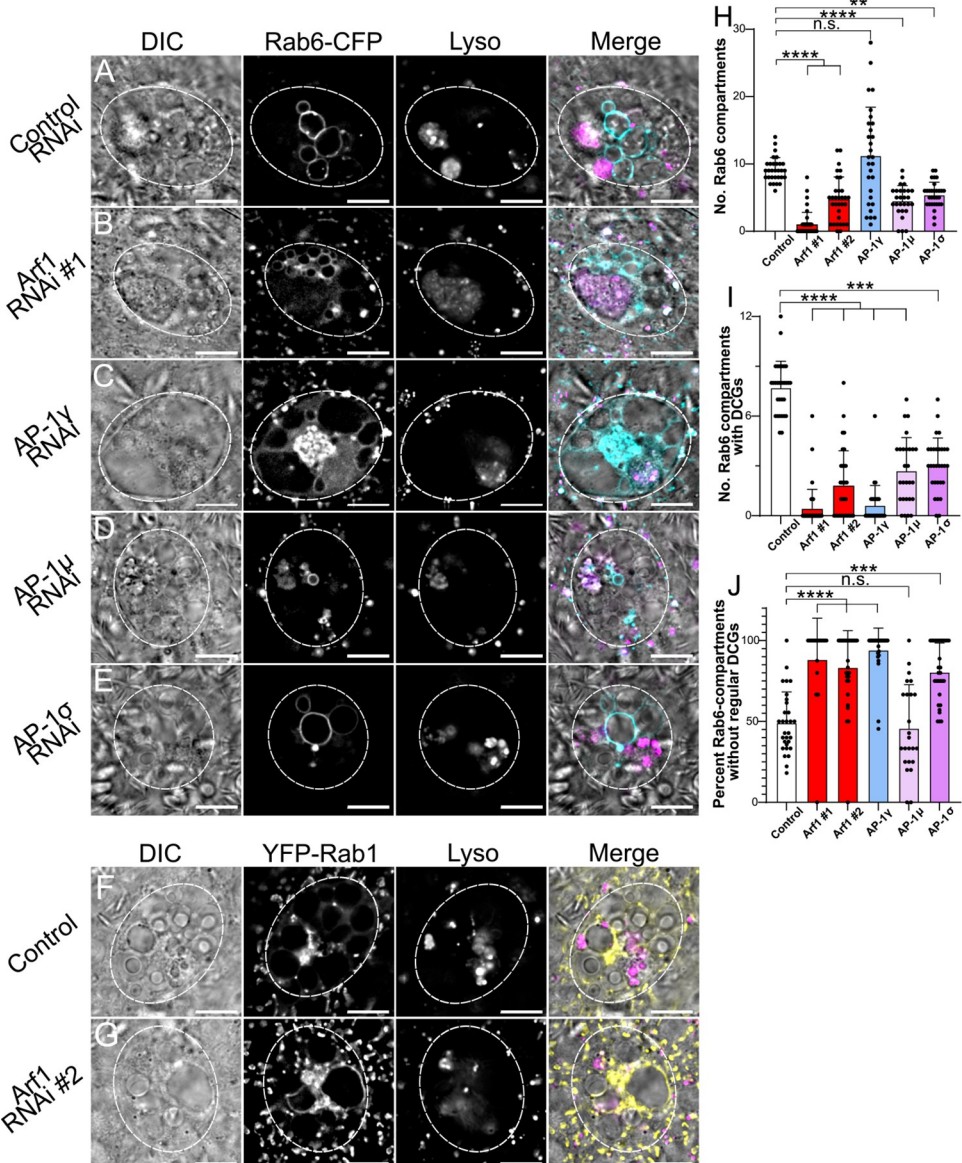

**Fig 6. *Arf1* and *AP-1* regulate Rab6-compartment identity and the maturation of DCG compartments.** (A-E) Representative images of SCs expressing the *CFP-Rab6* fusion gene from the endogenous *Rab* locus together with a control RNAi (A) or RNAis targeting *Arf1* (RNAi #1; B), *AP-1γ* (C), *AP-1μ* (D) or *AP-1σ* (E). Cellular organisation is assessed through DIC imaging, CFP-Rab6 fluorescence and Lysotracker Red fluorescence, and a merged image is presented for each cell. Note the number of CFP-Rab6-positive compartments is reduced in all knockdown backgrounds, except *AP-1γ*, where a central cluster of small Rab6-positive compartments is often also observed, and in all knockdowns, few labelled compartments contain DCGs. (F, G) Representative SCs expressing the *YFP-Rab1* fusion gene either alone (F) or together with an *Arf1* RNAi #2 (G). Note that in control *YFP-Rab1* SCs, no large non-acidic compartments are Rab1-positive. When SCs are subjected to *Arf1* knockdown, by contrast, YFP-Rab1 does mark several large non-acidic compartments which do not contain DCGs. (H) Bar chart showing number of CFP-Rab6-labelled compartments in control SCs and following knockdown of *Arf1* and *AP-1*. (I) Bar chart showing number of CFP-Rab6-compartments containing DCGs in these different genotypes. (J) Bar chart showing the percentage of CFP-Rab6-compartments, which fail to produce regularly shaped DCGs in these different genotypes. Approximate outlines of SCs are marked by dashed circles. Scale bars: 10 μm. For H-J, bars show mean ± SD; Control, n = 30; *Arf1* #1, n = 39; *Arf1* #2, n = 32; *AP-1γ*, n = 29; *AP-1μ*, n = 27; *AP-1σ*, n = 30. P<0.05: *, P<0.01: **, P<0.001: ***, P<0.0001: ****. Genotypes for images: (A) *w$^{1118}$; P{tub-GAL80$^{ts}$}, TI{TI}Rab6$^{CFP}$/P{ry$^{TRiP.HMS02827}$}; dsx-GAL4/+;* (B) *w$^{1118}$; P{tub-GAL80$^{ts}$}, TI{TI}Rab6$^{CFP}$/+; dsx-GAL4/P{Arf1$^{GD12522}$};* (C) *w$^{1118}$; P{tub-GAL80$^{ts}$}, TI{TI}Rab6$^{CFP}$/+; dsx-GAL4/P{AP-1γ$^{TRiP.JF02684}$};* (D) *w$^{1118}$; P{tub-GAL80$^{ts}$}, TI{TI}Rab6$^{CFP}$/+; dsx-GAL4/P{AP-1μ$^{GD14206}$};* (E) *w$^{1118}$; P{tub-GAL80$^{ts}$}, TI{TI}Rab6$^{CFP}$/P{AP-1σKK108869}; dsx-GAL4/+;* (F) *w1118; +; TI{TI}Rab1$^{CFP}$/+;* (G) *w$^{1118}$; P{tub-GAL80$^{ts}$}/P{Arf1$^{KK101396}$}; dsx-GAL4/TI{TI}Rab1$^{EYFP}$.*

Nevertheless, knockdown of *Arf1*, *AP-1γ*, *AP-1μ* and *AP-1σ* in SCs significantly reduced the total number of DCG compartments and the number of Rab11-positive compartments produced (Fig 5A–5G and S5D). Indeed, the most common phenotype for the strongest *Arf1* and *AP-1* knockdowns, *Arf1*-RNAi #1 and *AP-1γ*, was a complete deficiency of DCG compartments, consistent with our results with the GFP-GPI marker (Figs 4F and 5G). Interestingly, for *Arf1* knockdowns, when Rab11-compartments were formed, most of them contained DCGs. By contrast, for the *AP-1γ* knockdown, more cells produced Rab11-compartments (Fig 5F), but a lower proportion contained DCGs (S6B Fig). This suggests that following *Arf1* knockdown, DCG biogenesis can still take place in compartments that have undergone the Rab6 to Rab11 transition, while AP-1 may have other roles following the transition that are essential for any DCG biogenesis. Notably however, for all knockdowns a greater proportion of the DCGs produced were malformed or irregular, often being split into multiple fragments within a compartment (Fig 5H). This indicates that factors required for entirely normal DCG biogenesis or maturation are not present at appropriate levels in these compartments following either *Arf1* or *AP-1* knockdown.

All but one of the different knockdowns also reduced the number of Rab6-positive compartments. In particular, in the *Arf1*-RNAi #2 knockdown, only $1.0 \pm 1.2$ CFP-Rab6-labelled compartments were observed, while for *Arf1*-RNAi #1, several smaller Rab6-positive compartments were often formed (Figs 6A–6E and 6H, S5H). Interestingly, in both *Arf1* knockdowns, but not in *AP-1* subunit knockdowns, a significant proportion of large non-acidic, core-less compartments was not strongly labelled with CFP-Rab6 (S6D Fig), suggesting that these compartments had not matured to Rab6 identity. Indeed, when *Arf1* was knocked down in SCs expressing the *YFP-Rab1* fusion from the endogenous *Rab1* locus, a number of these large non-acidic, non-DCG compartments were strongly marked by Rab1; this feature was not observed in controls (compare Fig 6G and 6F). These Rab1-compartments are presumably stalled in a maturation step required to make Rab6-compartments, but they continue to enlarge within the *trans*-Golgi. This is either by expansion, because downstream maturation is blocked, or by fusion with smaller Rab1-positive compartments. We found it difficult to construct *Arf1* knockdown males expressing both the YFP-Rab1 and CFP-Rab6 markers in order to distinguish these two possibilities directly.

Whatever the explanation, it appears that *Arf1* knockdown leads to the production of large non-acidic, non-DCG compartments that are different from control and *AP-1 subunit* knockdown SCs, in that some of them do not carry Rab6, and several harbour Rab1 at their surface. This suggests that Arf1 plays an important role in an early stage of DCG compartment maturation, which is required for the Rab1 to Rab6 transition to take place. YFP-Rab2 has a related, but less extensive, Golgi distribution to YFP-Rab1 in control cells (S2 Fig). Interestingly, we found no evidence for Rab2 association with the enlarged compartments produced by *Arf1* knockdown (S5H Fig).

By contrast, although knockdown of some AP-1 subunits did reduce the number of large Rab6-positive compartments, AP-1 may not be essential for Rab6 recruitment to precursor secretory compartments formed at the *trans*-Golgi surface. Indeed, as well as often containing multiple large Rab6-compartments, the majority of *AP-1γ* knockdown SCs contained a large central cluster of unusually small Rab6-positive compartments (Fig 6C). This is consistent with the accumulation of newly generated, but immature, Rab6-compartments near to the *trans*-Golgi.

Consistent with Arf1 and AP-1 being directly or indirectly involved in the Rab6 to Rab11 transition during DCG compartment maturation, a smaller number and proportion of Rab6-positive compartments contained DCGs for all knockdowns compared to controls (Figs 6I and S6C), suggesting that *Arf1* and *AP-1* knockdowns suppress the Rab6 to Rab11 transition that

accompanies DCG biogenesis. Likewise, as observed in the YFP-Rab11 background, a significantly higher proportion of the DCGs that were produced were irregular and/or fragmented in most knockdowns (Fig 6J). The *AP-1γ* knockdown, which produced the strongest AP-1 knockdown phenotypes, was again particularly notable, because the average number of Rab6-positive compartments was not reduced compared to controls (Fig 6H). Since the number of Rab11-positive compartments was strongly reduced in this knockdown (Fig 5F), and, in most cells, none of the compartments contained DCGs (Fig 6I), this is consistent with AP-1 playing a key role in the Rab6 to Rab11 transition, and the hypothesis that this transition is required for DCG biogenesis.

In summary, homologues of Arf1 and the AP-1 subunits play critical roles in DCG biogenesis in SCs, as has also been reported in mammalian cells. Our data suggest that Arf1 may act earlier in the process than AP-1, since unlike AP-1, it seems to be required for large non-acidic compartments to transition to Rab6 identity. However, if Arf1 expression levels are sufficiently high to allow Rab6 compartments to form, many of these can mature into Rab11-positive DCG compartments. By contrast, based on our analysis of *AP-1γ* knockdown, AP-1 seems to have a particularly important involvement in the Rab6 to Rab11 transition and in subsequent maturation events that induce DCG formation. Therefore, the wide range of genetic and imaging tools available to analyse the large secretory compartments of SCs has allowed us to provide evidence that the roles of Arf1 and AP-1 in some of the processes leading to maturation of DCG progenitor compartments and formation of DCGs potentially differ.

## The Rab6 to Rab11 transition is required for DCG and exosome biogenesis

In light of the correlation between DCG biogenesis and the Rab6 to Rab11 transition, we hypothesised that this transition might be required for the assembly of DCGs. To test this, two independent RNAis were used to knockdown either *Rab6* or *Rab11* in SCs. This allowed us to determine for what processes in DCG biogenesis each Rab was required. Where both Rabs were found to be necessary for a process to occur, we concluded that the Rab6 to Rab11 transition was required to facilitate that process.

Inducing knockdown of either *Rab6* or *Rab11* in SCs expressing the GFP-GPI DCG marker led to a strong reduction in the number of DCGs present in SCs (Figs 7A–7D and S7A–S7C Fig). Indeed, the vast majority of cells expressing the most potent RNAis, *Rab6*-RNAi #1 and *Rab11*-RNAi #1, which have both been used to suppress Rab expression in previous studies [42,29], had no DCGs. Alongside this reduction in DCG-containing compartments, significantly fewer non-acidic compartments were present and many of those remaining were marked by diffuse GFP (Fig 7E and 7F). This mirrored our observations following the strongest knockdowns for *Arf1* and *AP-1* in the GFP-GPI background (Fig 4F–4H). Likewise, the proportion of acidic compartments containing fluorescent GFP increased following either *Rab6* or *Rab11* knockdowns (S7D Fig). This suggests that these knockdowns either increase trafficking of secretory compartments to the lysosomal pathway or disrupt lysosome activity. Overall, our results indicate that both Rab6 and Rab11 are required for DCG biogenesis.

Rab function can also be assessed by expressing constitutively active or dominant-negative mutant forms of these molecules under UAS control [43]. However, the effectiveness of these constructs is cell context-dependent, being affected by the relative level of mutant versus wild type gene expression and the mechanisms by which Rabs are recruited to their different subcellular sites of activity and subsequently inactivated.

Since Rab6 remains associated with maturing DCG compartments following the Rab6 to Rab11 transition (Figs 1C, 1I and 3), we hypothesised that it does not need to be inactivated for DCG biogenesis to take place. Indeed, a constitutively active, GTPase-defective form of

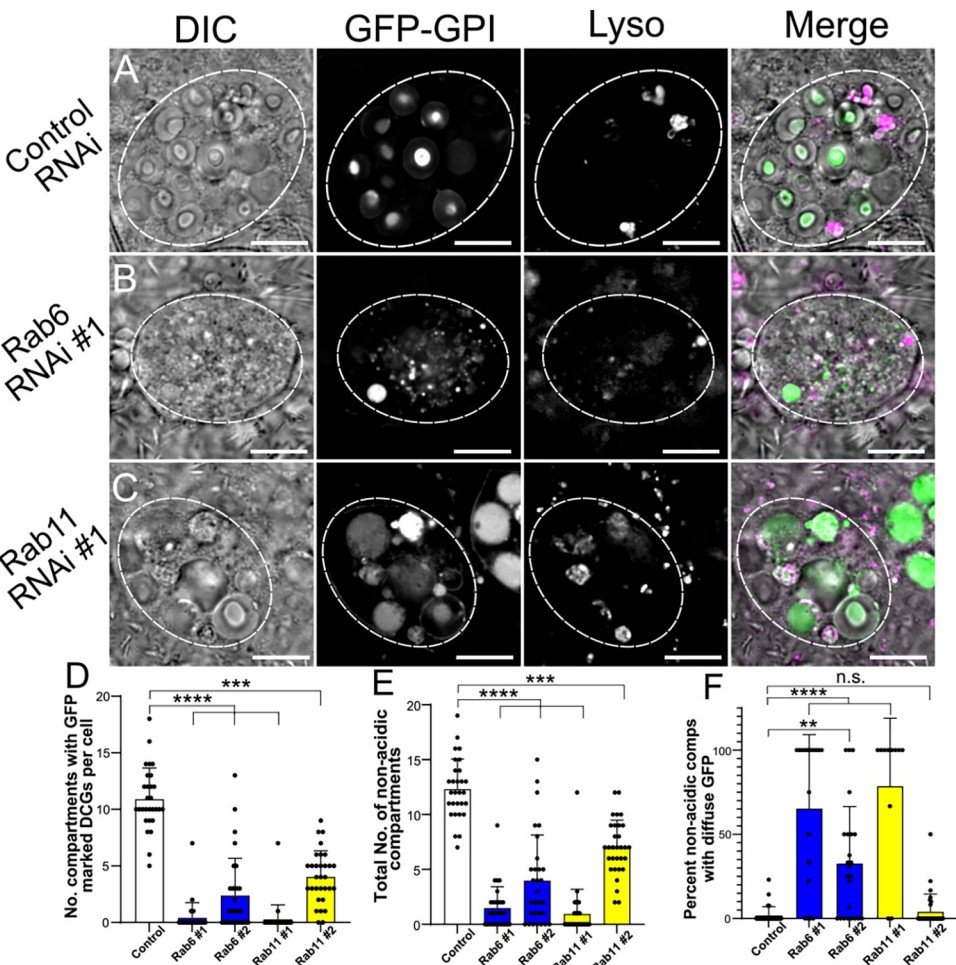

**Fig 7. Rab6 and Rab11 are both required for DCG biogenesis in SCs.** (A-C) Representative images of SCs expressing the DCG marker *GFP-GPI* with a control RNAi (A) or RNAis targeting *Rab6* (B) or *Rab11* (C). Cellular organisation is assessed by DIC imaging, GFP-GPI fluorescence and Lysotracker Red fluorescence, and a merged image for each cell. Note the reduction in large non-acidic compartments in these backgrounds with fewer containing DCGs and in some cases, the remaining compartments often filled with diffuse GFP. (D) Bar chart showing the number of compartments containing GFP-labelled DCGs in control SCs and following knockdown of *Rab6* and *Rab11*. (E) Bar chart showing the number of large non-acidic compartments in these different genotypes. (F) Bar chart showing the percentage of large non-acidic compartments with diffuse GFP-GPI present in these different genotypes. Approximate outlines of SCs are marked by dashed circles. Scale bars: 10 μm. For D-F, bars show mean ± SD; Control, n = 29; *Rab6* #1, n = 29; *Rab6* #2, n = 32; *Rab11* #1, n = 30; *Rab11* #2, n = 31. P<0.05: *, P<0.01: **, P<0.001: ***, P<0.0001: ****. Genotypes for images: (A) $w^{1118}$; P{tub-GAL80$^{ts}$}/P{ry$^{TRiP.HMS02827}$}; dsx-GAL4, P{UAS-GFP.GPI}/+; (B) $w^{1118}$; P{tub-GAL80$^{ts}$}/+; dsx-GAL4, P{UAS-GFP.GPI}/P{Rab6$^{TRiP.HMS01486}$}; (C) $w^{1118}$; P{tub-GAL80$^{ts}$}/+; dsx-GAL4, P{UAS-GFP.GPI}/P{Rab11$^{TRiP.JF02812}$}.

Rab6, Rab6-CA, tagged with YFP, which has been employed successfully in previous studies [44], behaved almost identically to wild type YFP-Rab6, when expressed only in adult SCs under UAS/GAL80$^{ts}$ control (S8A and S8B Fig). Like wild type YFP-Rab6, it mostly co-localised with CFP-Rab6 expressed from the endogenous *Rab6* locus (S8D and S8E Fig). However, it labelled slightly fewer large non-acidic compartments and DCG compartments than UAS-regulated wild type YFP-Rab6 (S8H, S8I, S8K and S8L Fig). The total number of DCG compartments was unaffected by this mutant protein (S8J and S8M Fig), which could still dissociate from the most mature DCG compartments (S8B and S8B Fig). This suggests either that

inactivation of Rab6 is not critical for release of this protein from maturing DCG compartments or that in the context of the SC, the YFP-Rab6-CA protein does not behave as a constitutively active protein.

Expression of YFP-labelled dominant-negative Rab6, YFP-Rab6-DN, blocked the formation of all DCG compartments and led to the formation of many small compartments that lacked a DCG (S8C and S8H–S8M Fig), a phenotype similar to *Rab6*-RNAi #1 knockdown cells (Fig 7B). However, a large proportion of the mutant protein was inappropriately sequestered into the two SC nuclei, which might lead to effects that are unrelated to Rab6 inhibition. Furthermore, it was also difficult to assess whether the protein was associated with the limiting membranes of compartments, because very little cytosol remained in the cell (S8C Fig).

Interestingly, when YFP-Rab6-DN was produced in a *CFP-Rab6*-expressing background, only a few SCs had a similar phenotype to when YFP-Rab6-DN was expressed alone (S8G–S8M Fig). In these cells, neither wild type CFP-Rab6 nor YFP-Rab6-DN associated with the multiple compartments that were formed and these lacked DCGs. However, the majority of cells expressing these two fusion proteins contained a variable number of DCG compartments, with most of these compartments co-labelled with YFP and CFP, and just a few compartments marked by only one Rab6 fusion protein (S8F and S8H–S8M Fig). These results suggest that the Rab6-DN mutant may act as a dominant-negative in some cells, which then fail to form DCGs (S8G Fig). However, in other cells that perhaps express the mutant protein at lower levels, it does not prevent wild type Rab6 from associating with secretory compartment membranes and promoting formation of DCGs (S8F Fig). Indeed, in this scenario, Rab6-DN also appears to bind to the limiting membrane of these compartments and therefore, does not appear to be acting in a classical dominant-negative fashion.

When we expressed in SCs either a YFP-Rab11-CA or YFP-Rab11-DN construct, both of which have been used successfully in previous studies [45,46], the levels of fluorescence detected were relatively low compared to other Rab fusion proteins, including all the Rab fusions expressed from their endogenous gene loci (S9 Fig). Neither YFP-Rab11-CA nor YFP-Rab11-DN were localised on DCG compartment membranes (S9H Fig), and they had no effect on the total number of DCG compartments in SCs (S9I Fig). When co-expressed with CFP-Rab6, the latter's localisation was unaffected (S9E and S9F Fig). We conclude that in contrast to overexpressing wild type YFP-Rab11 (S9A Fig), these mutant proteins can only be expressed in SCs at very low levels, which do not significantly affect Rab11 activity. This conclusion was confirmed for YFP-Rab11-DN by co-expression with YFP-Rab11 made from the endogenous *Rab11* locus. In the resulting cells, YFP, presumably associated with the wild type protein, was localised to the limiting membrane of DCG compartments (S9G Fig). Interestingly, it also concentrated in small subdomains at the outer surface of some of these compartments. We speculate that these subdomains may represent small Rab11-positive compartments that interact with the large DCG compartments to deliver and/or remove cargos, and whose activity may be subtly affected by the YFP-Rab11-DN protein.

To better understand the functions of Rab6 and Rab11, we looked in more detail at the effects of *Rab6* and *Rab11* knockdown, using the *CFP-Rab6* and *YFP-Rab11* backgrounds. Just as seen in the GFP-GPI background, all knockdowns significantly reduced the number of Rab6- and Rab11-marked compartments forming DCGs with one exception. Many cells expressing *Rab11*-RNAi #2 contained an increased number of Rab6-compartments, leading to the number of DCG-containing Rab6-compartments being unaffected (Figs 8A–8H and 8L and S10). Interestingly, the loss of DCGs in these endogenously tagged *Rab* gene backgrounds was not as severe as the loss observed in the GFP-GPI background. This mirrored our findings with *Arf1* and *AP-1* subunit knockdowns, where we concluded that knockdown of these molecules was reduced in these Rab marker lines. This conclusion is supported by the fact that a

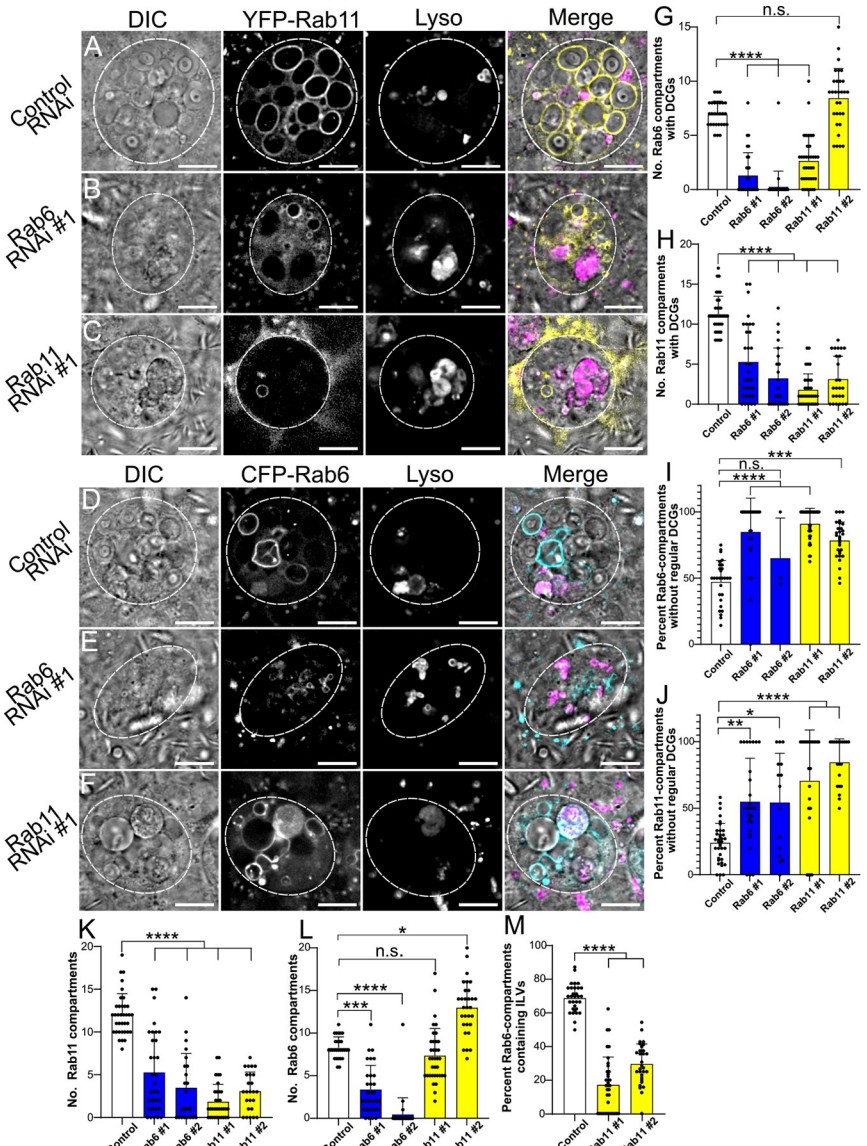

**Fig 8. The Rab6 to Rab11 transition on secretory compartments controls exosome as well as DCG biogenesis in SCs.** (A-C) Representative images of SCs expressing the *YFP-Rab11* fusion gene from the endogenous *Rab* locus with a control RNAi (A) or RNAis targeting *Rab6* (B) or *Rab11* (C). (D-F) Representative images of SCs expressing the *CFP-Rab6* fusion gene from the endogenous *Rab* locus with a control RNAi (D) or RNAis targeting *Rab6* (E) or *Rab11* (F). Cellular organisation in all genotypes is assessed through DIC imaging, tagged Rab fluorescence and Lysotracker Red fluorescence, as well as a merged image for each cell. Note that when *Rab6* is knocked down, there are reduced numbers of Rab11-compartments and fewer of those that remain contain normal DCGs (H, J, K). By contrast, *Rab11* knockdown does not reduce the number of Rab6-positive compartments, but fewer of these compartments contain normal DCGs or ILVs (G, I, L, M). Also, note that some Rab11 fusion gene fluorescence is still visible even after knockdown of *Rab11* (C), and similarly for CFP-Rab6 in the *Rab6* knockdown (E). (G) Bar chart showing the number of CFP-Rab6-compartments containing DCGs in control SCs and following *Rab6* and *Rab11* knockdown. (H) Bar chart showing the number of YFP-Rab11-compartments containing DCGs in these different genotypes. (I) Bar chart showing the percentage of CFP-Rab6 compartments which fail to produce regular DCGs in these different genotypes. (J) Bar chart showing the percentage of YFP-Rab11 compartments which fail to produce regular DCGs in these different genotypes. (K) Bar chart showing the number of YFP-Rab11-compartments in these different genotypes. (L) Bar chart showing the number of CFP-Rab6 compartments in these different genotypes. (M) Bar chart showing the percentage of CFP-Rab6 compartments which contain CFP-Rab6-labelled ILVs following knockdown of *Rab11* in SCs versus controls. Approximate outlines of SCs are marked by dashed circles. Scale bars: 10 μm. For G-M, bars show mean ± SD. For G, I and L, Control, n = 29; *Rab6* #1, n = 30; *Rab6* #2, n = 32; *Rab11* #1, n = 36; *Rab11* #2, n = 27. For H, J and K, Control, n = 36; *Rab6* #1, n = 32; *Rab6* #2, n = 23; *Rab11* #1, n = 33; *Rab11* #2, n = 25. For M, Control,

n = 31; *Rab11* #1, n = 36; *Rab11* #2, n = 30. P<0.05: *, P<0.01: **, P<0.001: ***, P<0.0001: ****. Genotypes for images: (A) *w^1118^; P{tub-GAL80^ts^}/P{ry^TRiP.HMS02827^}; dsx-GAL4, TI{TI}Rab11^EYFP^/+;* (B) *w^1118^; P{tub-GAL80^ts^}/+; dsx-GAL4, TI{TI}Rab11^EYFP^/P{Rab6^TRiP.HMS01486^};* (C) *w^1118^; P{tub-GAL80^ts^}/+; dsx-GAL4, TI{TI}Rab11^EYFP^/P{Rab11^TRiP.JF02812^};* (D) w^1118^; P{tub-GAL80^ts^}, TI{TI}Rab6^CFP^/P{ry^TRiP.HMS02827^}; dsx-GAL4/+; (E) *w^1118^; P{tub-GAL80^ts^}, TI{TI}Rab6^CFP^/ +; dsx-GAL4/P{Rab6^TRiP.HMS01486^};* (F) *w^1118^; P{tub-GAL80^ts^}, TI{TI}Rab6^CFP^/+; dsx-GAL4/P{Rab11^TRiP.JF02812^}.*

small number of compartments marked by YFP-Rab11 and CFP-Rab6 were still present even after knockdown of the corresponding *Rab*, with *Rab11*-RNAi #2 being less effective than *Rab11*-RNAi #1 (Figs 8C and S10C), and Rab6-RNAi #1 permitting the formation of a few small Rab6-positive compartments in some cells (Fig 8E).

We next investigated whether those DCGs that formed after *Rab* knockdown matured normally, by analysing the number of compartments that contained either immature/irregular DCGs or no DCG. We found that knockdown of *Rab6* or *Rab11* significantly increased the proportion of Rab6- and Rab11-compartments lacking a mature, regular DCG, both in CFP-Rab6- and YFP-Rab11-labelled compartments (Fig 8I and 8J). Indeed, many cells did not contain any normal DCGs. This included knockdown with leaky *Rab11*-RNAi #2 (Fig 8G), where approximately 75% of Rab6-compartments were either core-less or contained irregular DCGs (Fig 8I). It also applied to *Rab6*-RNAi #2, even though many SCs could not be scored, because they lacked any Rab6- or Rab11-labelled compartments (Fig 8K and 8L), Overall, these results mirror those for *Arf1* and *AP-1* knockdowns, indicating that both Rab6 and Rab11 are required for DCG biogenesis, since knockdown of either *Rab* reduces the total number of DCGs (Figs 7D, 8G and 8H) and decreases the proportion of large secretory compartments that can form a mature regularly shaped DCG (Fig 8I and 8J).

To further assess whether the compartment transition from Rab6- to Rab11-positive identity is required for DCG biogenesis, we also looked at the effect of *Rab6* and *Rab11* knockdown on secretory compartment identity. Following knockdown of *Rab6* in a *YFP-Rab11* background, we observed a very large reduction in the number of Rab11-positive compartments, with an approximately 75% decline in these compartments relative to controls and many cases where no Rab11-compartments were formed (Fig 8K). Importantly, this was accompanied by the large decline in mature DCGs discussed earlier (Fig 8H and 8J). In contrast, knockdown of *Rab11* in a *CFP-Rab6* background also prevented normal DCG formation, but had no discernible effect on the number of Rab6-positive compartments or even increased their number (Fig 8L). These results are consistent with our finding that a Rab6 to Rab11 transition occurs as secretory compartments mature, and that this is required for the production of DCGs.

Finally, having shown that Rab11 is required for DCG formation, we examined its role in biogenesis of exosomes produced in Rab11-compartments, collectively termed Rab11-exosomes. As shown earlier, whilst imaging control samples in the *CFP-Rab6* background, we had noted the very strong correlation between the presence of DCGs and Rab6-positive ILVs within compartments, with these ILVs almost exclusively appearing in DCG compartments (Fig 1M). These ILVs are destined to be secreted as Rab11-exosomes. Since time-lapse imaging had shown that ILVs and DCGs appear in compartments within minutes of each other, we tested the hypothesis that the Rab6 to Rab11 transition is also the trigger for ILV (and consequently Rab11-exosome) biogenesis. Following knockdown of *Rab11* in a *Rab6-CFP* background, a significantly smaller proportion of Rab6-positive compartments produced Rab6-labelled ILVs than in controls (Fig 8M). As with DCG formation, this result indicates that the recruitment of Rab11 to the membrane of compartments derived from the *trans*-Golgi network is an important step for biogenesis of Rab11-exosomes.

## Discussion

Genetic dissection of DCG biogenesis in the regulated secretory pathway has been restricted by availability of *in vivo* models and the limited possibilities to employ fluorescence and real-time imaging to study the processes involved. Here, we employ SCs of the *Drosophila* male accessory gland to overcome these hurdles. We demonstrate that some of the best character-ised regulators of DCG biogenesis in mammals are also involved in DCG formation in SCs and show that a series of Rab transitions involving Rab1, Rab6 and Rab11 precede DCG for-mation. These transitions suggest a critical interaction between the secretory and recycling endosomal pathways, which is controlled by trafficking regulators like Arf1 and AP-1 (see model in Fig 9).

### Rab1 and Rab6 mark secretory compartments prior to DCG formation

By expressing fluorescent Rab proteins from the endogenous *Rab* locus in SCs we were able to produce detailed time-lapse videos of maturing secretory compartments. These showed that both Rab1 and Rab6 are present on compartments produced at the *trans*-face of the Golgi at successive stages of maturation, with small Rab1-positive compartments transitioning into larger Rab6-positive compartments over the course of approximately 40 minutes (Figs 2, 9 and S3). Our time-lapse videos also provided insights into the mechanisms of secretory compart-ment maturation, showing that multiple smaller Rab1-/Rab6-labelled compartments can fuse to create the larger Rab6-compartments in SCs. After the Rab1-Rab6 transition, when Rab1-YFP signal is no longer visible on the limiting membrane of compartments, a DCG can then form (Fig 2). This conclusively demonstrates that these Rab1/Rab6-marked compart-ments are the same structures that go on to form DCGs. Our results fit well with previous find-ings that Rab1 drives secretory granule maturation in *Drosophila* salivary glands and with observations that Rab1 and Rab6 are located at the periphery of immature granule-forming compartments [29,30], although in salivary glands, Rab1 may remain associated with more mature compartments.

### The Rab6 to Rab11 transition accompanies DCG biogenesis and is modulated by known DCG regulators

Through further time-lapse imaging, we found that another Rab transition takes place on the surface of secretory compartments prior to DCG biogenesis. Rab6 on the surface of compart-ments is gradually replaced by Rab11 over the course of many hours, with DCG biogenesis occurring minutes after the beginning of this transition (eg. Fig 3). From the time-lapse videos it appeared that the entire process of DCG formation was typically complete within 20 minutes.

We also showed, through SC-specific knockdown of *Arf1* and each of the *AP-1* subunits, that the roles of these molecules in DCG biogenesis are conserved in SCs. Both Arf1 and AP-1 are evolutionarily conserved trafficking regulators that are known to be required for matura-tion of DCG compartments [7–9]. Knockdown of *Arf1* and individual *AP-1* subunits, using UAS-RNAi lines that had been successfully employed in previous studies, produced a range of phenotypes, even for a single RNAi, presumably because of different levels of RNAi expression in individual SCs. Furthermore, there were differences in phenotypes induced by different RNAis, particularly when comparing results for the three AP-1 subunits, consistent with dif-ferent levels of subunit knockdown. However, for all *Arf1* and *AP-1* knockdowns in SCs expressing a range of markers, DCG formation was significantly reduced. Indeed, for the most potent RNAis, *Arf1*-RNAi #1 and *AP-1γ*-RNAi, in GFP-GPI-expressing SCs, DCGs were

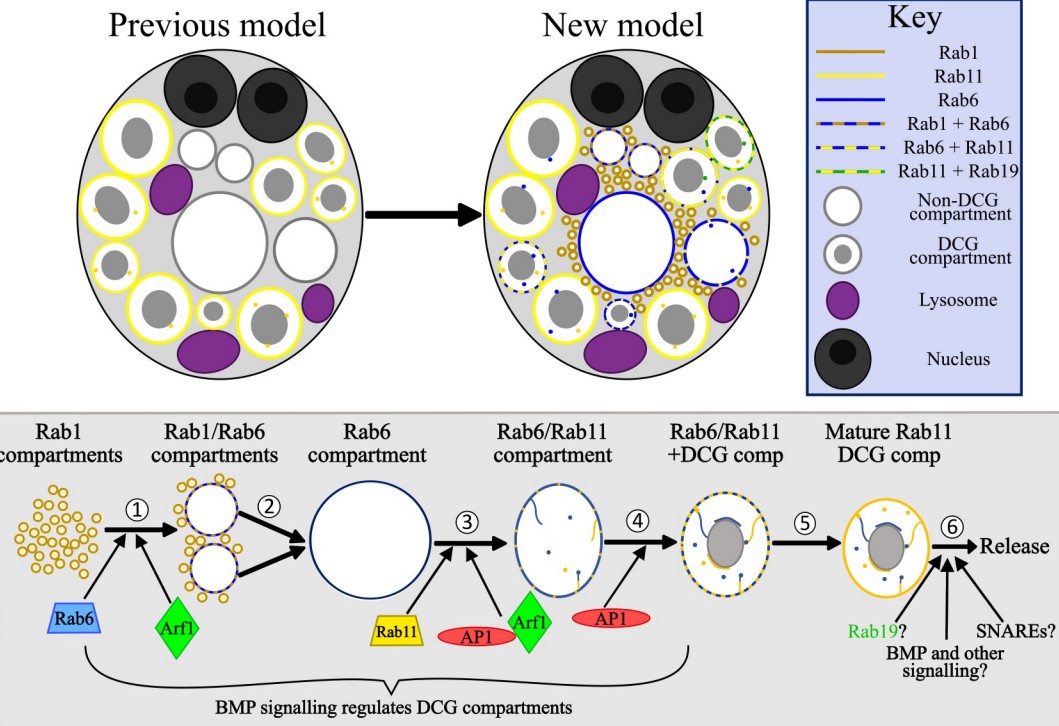

**Fig 9. Model for the regulation of DCG compartment biogenesis in SCs.** (A) A schematic illustrating our previous and updated model of SC secretory and endosomal compartment organisation. Whereas previously it was recognised that DCG compartments in SCs were labelled by Rab11, we have now shown that Rab6 and Rab19 also mark large secretory compartments and can co-label compartments alongside Rab11. We have also demonstrated that Rab1 marks a population of smaller compartments near the cell centre and can colocalise with Rab6 on the surface of growing compartments. Finally, as well as the Rab11-positive ILVs which were recognised beforehand, we have also described the existence of ILVs marked by Rab6 and Rab19, which can be found in compartments labelled by Rab11, and will be secreted as Rab11-exosomes. (B) Schematic outlining the genetic regulation of secretory compartment maturation and DCG biogenesis in SCs. Our results have highlighted at least 6 discrete stages which occur during secretory compartment maturation. (1) In the earliest stage, small Rab1-compartments fuse together and recruit Rab6 to their limiting membrane, creating enlarged Rab1/Rab6-positive compartments. (2) These compartments continue to grow, at least in part via fusion events, until they eventually lose all Rab1. They are then marked solely by Rab6, and contain neither DCGs nor ILVs. The Rab1 to Rab6 transition is regulated by Arf1 and Rab6 recruitment is required to progress to later maturation steps. (3) Soon after formation, Rab6-positive compartments contract in size. They subsequently recruit Rab11 to their limiting membrane, inducing the formation of ILVs, some of which appear to coalesce into the long ILV chains we have observed (internal lines in compartments). The recruitment of Rab11 is regulated by Arf1 and the AP-1 complex, without which most Rab11-compartments fail to form. Any that do form usually do not mature normally. (4) As Rab11 continues to be recruited to membranes, DCG biogenesis occurs within compartments. BMP signalling regulates the rate of DCG-compartment biogenesis, indicating that BMP acts at one or more points upstream of this event [32]. Additionally, AP-1 regulates normal DCG biogenesis. (5) Over several more hours, Rab6 is fully shed from the limiting membrane, leaving a mature secretory compartment which contains a DCG and a mix of ILVs. (6) Matured compartments are eventually secreted following fusion with the plasma membrane. Secretory compartment release is regulated by BMP signalling [32], but the other factors involved remain unclear. Specific SNARE proteins are likely required for fusion to the plasma membrane, and compartments may undergo further maturation steps, possibly regulated by factors such as Rab19 and additional signalling pathways.

absent in almost all SCs (Fig 4F). Additionally, experiments in *CFP-Rab6* and *YFP-Rab11* backgrounds showed that knockdown of *Arf1* and *AP-1* subunits significantly decreased the number of Rab11-positive compartments in all cases (Fig 5) and of Rab6-positive compartments for *Arf1* knockdown (Fig 6). Of those compartments that remained in these knockdown cells, a significantly larger proportion failed to form regular DCGs.

Despite the variability of phenotypes produced for each RNAi, it was possible to discern some differences in the effects of *Arf1* and *AP-1* knockdown. Based on our analysis, we suggest

that Arf1 and AP-1 contribute to the process of DCG biogenesis at three different stages (Fig 9). Firstly, Arf1 seems to be involved in a step required for the Rab1 to Rab6 transition stage, thereby explaining the presence of large Rab1-positive, non-DCG compartments in *Arf1* knockdowns, and the fact that some large, non-DCG compartments are not marked by Rab6 (S6D Fig). This phenotype is not observed for *AP-1 subunit* knockdowns. Secondly, since Rab11-compartments and DCG biogenesis are also strongly reduced in *Arf1* and *AP-1* knockdowns, we propose that their protein products are required either directly or indirectly during the Rab6 to Rab11 transition and associated DCG formation. This effect is perhaps most obvious for the *AP-1γ* knockdown, where Rab6 compartments are formed, but relatively few transition to Rab11-compartments (Figs 5F and 6H). Thirdly, even if this transition takes place and DCGs form, likely because knockdown does not completely eliminate all target gene expression, *AP-1* in particular seems to be required for the establishment of regular DCG morphology (eg. *AP-1γ* knockdown; Fig 6H–6J). One explanation for these latter two roles in DCG biogenesis is that Arf1 and AP-1 are directly or indirectly involved in the fusion and release of vesicles to and from Rab6-/Rab11-positive compartments, thereby delivering materials that are needed for DCG formation and removing those that are not (Fig 9). Regarding an indirect role for Arf1, we observed that Rab11-compartments that do form in *Arf1* knockdown cells frequently make DCGs (Fig 5F and 5G). Arf1 is known to be required to recruit AP-1 to the Golgi [7]. It is therefore possible in our *Arf1* knockdown experiments that if there is enough Arf1 to form a large Rab6-compartment, this compartment will mature to a Rab11-positive DCG compartment, because Arf1 has recruited sufficient levels of downstream regulators like AP-1 during the early stages of the maturation process.

## The Rab6 to Rab11 transition is required for DCG biogenesis

Knockdown of either *Rab6* or *Rab11* in a GFP-GPI background, using the more potent of the two RNAis tested for each gene, effectively eliminated DCG formation in SCs, demonstrating that both Rabs are required for granule biogenesis (Fig 7). We also expressed constitutively active and dominant-negative forms of these Rabs in SCs to try to further investigate how these Rabs interact. However, these mutant constructs provided limited further insights, partly because some were only expressed at very low levels, and partly because they did not appear to interfere with Rab function in the predicted fashion in SCs (S8 and S9 Figs).

It was particularly significant that *Rab11* knockdown inhibited DCG formation, since this is consistent with our observations that the start of DCG biogenesis occurs after the beginning of the Rab6 to Rab11 transition. The critical role for *Rab11* at this transition stage was supported by the observation that the numbers of Rab6-positive precursor compartments were not reduced by *Rab11* knockdown (Fig 8L). Together with the observation that DCG biogenesis occurs early in the Rab6 to Rab11 transition, these results strongly suggest that the switch to Rab11 identity acts as the regulatory trigger for DCG biogenesis in SCs. Since Rab6 normally remains associated with the limiting membrane of less mature DCG compartments, it seems that it may not need to be inactivated for this transition to take place.

Rab11 is frequently associated with recycling endosomes [47,48]. However, it is also associated with secretory granules, such as Epidermal Lamellar Granules and the secretory granules of *Drosophila* salivary glands [49,29]. Likewise, Rab11 is also known to contribute to insulin granule exocytosis in pancreatic β cells [15], as well as to the trafficking and release of secretory granules in the fungus, *Aspergillus nidulans* [50]. It therefore seems likely that in these other cells, the association with Rab11 has a role to play in the proper maturation of the secretory compartments they produce and in DCG biogenesis, as we have found in SCs.

One important unanswered question is whether the Rab6 to Rab11 transition is associated with delivery of cargos specifically from Rab11-positive recycling endosomes that fuse with the large Rab6-compartments and promote DCG biogenesis. Our data suggest that the GFP-GPI marker enters DCG precursor compartments before the Rab6 to Rab11 transition, because knockdowns that block this transition lead to accumulation of diffuse GFP-GPI in the resulting non-acidic compartments that are formed. However, recycling endosomes are known to be more acidic than the secretory compartments that form at the surface of the *trans*-Golgi. Therefore, one possibility is that the recycling endosomal system delivers the V-ATPase proton pump or more acidic luminal contents during the Rab6 to Rab11 transition, which might lead to the pH changes in secretory compartments that are known to drive DCG assembly [10].

## Rab11-exosome and DCG biogenesis may be interdependent processes controlled by the Rab6 to Rab11 transition

Because of the unique biology of the SC system, we were able to examine additional aspects of DCG compartment regulation that we would not easily be able to investigate in other available models. One key feature was the association of ILVs with the developing DCG inside the maturing compartment. Imaging of control cells in a *CFP-Rab6* background showed that ILVs are almost exclusively found in compartments with DCGs (Fig 1M). Furthermore, consistent with previous observations (Fan et al., 2020), time-lapse imaging and single wide-field fluorescence micrographs demonstrated that ILVs marked by Rab19-YFP, Rab11-YFP and Rab6-CFP were closely associated with DCGs. Rab11 is an established marker of exosomes secreted from compartments marked by this recycling endosomal Rab [33], but both Rab19 [51,52] and Rab6 [53,54] have also been reported to be secreted in mammalian extracellular vesicles, suggesting that exosomes labelled by these Rabs are not uniquely produced by SCs.

Extended lengths of many DCG boundaries were clearly marked by Rab-labelled ILVs and the fluorescent Rab-signal revealed long chains of ILVs running from the limiting membranes of compartments to DCGs and then along DCG boundaries. Indeed, it was often possible to discern the position and shape of DCGs through just the distribution of fluorescent ILVs. In addition, we also found that knockdown of *Rab11* significantly reduced the formation of Rab6-positive ILVs, which normally contribute to the population of Rab11-exosomes originating from SC Rab11-compartments, thereby demonstrating that Rab11 is required for the biogenesis or stabilisation of these ILVs as well as for DCG formation (Fig 9). These results indicate that ILV and DCG biogenesis share common regulatory mechanisms.

A previous study has suggested that knockdown of some components of the core ESCRT complexes disrupts DCG biogenesis in SCs [34]. Importantly, however, not all *ESCRT* knockdowns have the same effect, suggesting that the formation of ILVs itself may not be the essential event providing this link. Previous studies in SCs have shown that vesicle-associated GAPDH activity regulates changes in ILV clustering which are closely linked with DCG morphology [38]. Together with these earlier findings, our results highlight the link between ILVs and DCGs and emphasise the need to understand the role of GAPDH and other membrane-associated molecules in regulating this interaction. Furthermore, they suggest that the conserved trafficking pathway that makes Rab11-exosomes [33,34] may also be critical in producing DCGs in secretory cells.

Finally, both DCG biogenesis and secretion in SCs is regulated by autocrine BMP signalling mediated by the BMP ligand, which is packaged into the DCGs, providing a mechanism for accelerating DCG production when secretion rates are high [32]. It will now be interesting to determine which maturation events in DCG biogenesis are controlled by BMP signalling

(Fig 9), and whether additional intracellular signalling cascades can modulate this process in other ways.

## Materials and methods

### Fly stocks

UAS-RNAi lines were sourced from the Bloomington *Drosophila* Stock Centre TRiP collection (BDSC; [55]) and Vienna *Drosophila* Resource Centre shRNA, GD and KK libraries (VDRC; [56]): *rosy-RNAi* as a control (BDSC; 44106; HMS02827; [34]); *Arf1-RNAi* #1 (VDRC; 23082; [40]) and #2 (VDRC; 103572; [40]); *AP-1γ-RNAi* (BDSC; 27533; JF02684; [41]); *AP-1μ-RNAi* (VDRC; 24017; [57]); *AP-1σ-RNAi* (VDRC; 107322); *Rab6-RNAi* #1 (BDSC; 35744; HMS01486; [42]) and #2 (BDSC; 27490; JF02640; [58]); *Rab11-RNAi* #1 (BDSC; 27730; JF02812; [29]) and #2 (VDRC; 108382; [59,29]). UAS-driven YFP-labelled versions of wild type and mutant Rab6 and Rab11 were also employed [43]: *UAS-YFP-Rab6* (BDSC; 23251); *UAS-YFP-Rab6$^{Q71A}$* (CA; BDSC; 9776); *UAS-YFP-Rab6$^{T26N}$* (DN; BDSC; 23250); *UAS-YFP-Rab11* (BDSC; 50782); *UAS-YFP-Rab11$^{Q70L}$* (CA; BDSC; 50783); *UAS-YFP-Rab11$^{S25N}$* (DN; BDSC; 23261). We additionally used the following endogenously tagged fluorescent *Rab* lines: *YFP-Rab11, YFP-Rab1, YFP-Rab19* [36] and *CFP-Rab6*, provided by S. Eaton and F. Karch; *UAS-GFP-GPI* [60,32]. The *CFP-Rab6, YFP-Rab11* and *UAS-GFP-GPI* lines were combined with *tub-GAL80$^{ts}$* (BDSC 7108) and *dsx-GAL4* (provided by S. Goodwin) to produce *Drosophila* lines with one of these three fluorescent markers, as well as the GAL4-GAL80$^{ts}$ machinery that allows SC-specific temperature-inducible expression of UAS-transgenes.

### Fly culture and handling

Flies were cultured on standard cornmeal agar food, using a 12-hour light/dark cycle. Flies carrying UAS-transgenes were crossed with flies containing the *dsx*-Gal4/tub-Gal80$^{ts}$ driver system as well as one of the fluorescent *Rab* genes or *UAS-GFP-GPI* and kept at 25˚C. Virgin male offspring from this cross were collected on the same day they eclosed and were transferred to 29˚C for 6 days to trigger transgene expression. In experiments where no UAS-transgene was expressed and only endogenously tagged genes were employed, the same timings and temperatures were used, except for time-lapse experiments where incubation at 29˚C varied from 5–7 days post-eclosion.

### Imaging, deconvolution and time-lapse movies

To visualise SC organisation, accessory glands were dissected into cold PBS, incubated with 500nM Lysotracker Red DN-99 (Invitrogen, L7528) for 5 minutes on ice and washed again with cold PBS. Finally, these *ex vivo*-prepared glands were mounted between two coverslips (thickness No. 1.5H, Marienfeld-Superior) in a small drop of PBS. Cover slips were placed into a custom-made metal mount for support. Excess PBS was drawn off using filter paper until glands were slightly flattened between the two coverslips.

Live SCs were imaged using the DeltaVision Elite system from Olympus AppliedPrecision, an inverted wide-field fluorescence microscope that can perform both fluorescence and differential interference contrast (DIC) microscopy. Cells were viewed at 1000X magnification using a 100x/1.40 oil emersion objective lens without auxiliary zoom lenses. CFP, YFP, GFP, Alexa Fluor 647 and LysoTracker Red were imaged using the following excitation and emission filters respectively: CFP, 438/24 and 475/24, YFP 513/17 and 548/22; GFP, 475/28 and 525/48; mCherry, 575/25 and 625/45; and Alexa Fluor 647, Cy5 632/22 and 676/34. An EMCCD Evolve-512 camera was used to capture images. Three SCs were imaged and analysed

from each accessory gland. For knockdown experiments, accessory glands from ≥10 individual males were imaged during 3 to 6 separate imaging sessions. To visualise the full 3D structure of SCs, Z-stacks were generated with 0.3 μm spacing between slices. The only exception to this was during the time-lapse imaging of the Rab6 to Rab11 transition for which only a single representative Z-plane was imaged in order to minimise bleaching and phototoxicity over the 6-8-hour experiments. Acquisition frequency during time-lapses varied between 4 minutes and 14 minutes in individual experiments. The SoftWoRx software was used to deconvolve Z-stacks to improve the signal:noise ratio in images prior to analysis.

## Fixation and immunostaining of accessory glands

To produce fixed samples, accessory glands were dissected as normal and then fixed by incubating them in 4% w/v paraformaldehyde in 1× PBS for 20 minutes. This and all other steps were conducted at room temperature unless otherwise stated. Immunostaining of fixed glands then proceeded as described in Corrigan et al., 2014 [31]. Following fixation, glands were washed in 1x PBS with 0.3% v/v Triton X-100 (Sigma-Aldrich), hereafter PBST, for 20 minutes. Glands were then transferred to PBST with 10% v/v Goat serum (Sigma-Aldrich), hereafter PBSTG, for 30 minutes. Glands were then incubated overnight at 4°C with Rabbit anti-GM130 antibody (Abcam #ab30637) diluted 1:100 in PBSTG. After this, the glands were washed six times for 10 minutes in PBST and then incubated with an anti-rabbit secondary antibody conjugated to Alexa Fluor 647 (Life Technologies #A-21245) diluted 1:400 in PBSTG for 2 hours. Stained glands were then washed five times in PBST, for 10 minutes each, with a final wash in 1× PBS for 10 minutes prior to mounting in Vectashield (Vector Laboratories) for wide-field imaging.

## Analysis and parameters

Deconvolved images were analysed in FIJI/ImageJ. To determine the number of compartments marked by a specific *Rab* gene-trap, every compartment was counted that was >1 μm in diameter at its widest point and displayed fluorescent signal that was greater than adjacent background cytosolic signal specifically at its limiting membrane. In cases where this was not clear, the transect tool on ImageJ was used to determine whether there was a peak in fluorescent signal at the limiting membrane. The fluorescent signal typically extended all around a compartment, but in the rare examples where this was not the case, the compartment was scored as positive if at least 25% of its perimeter was labelled. This excluded the Rab19 labelling of microdomains on some non-acidic compartments. To determine the number of DCGs in SCs, the DIC channel was used across all genotypes and was supplemented with the GFP channel in GFP-GPI backgrounds. The DIC channel was also used to assess the morphology of DCGs. The vast majority of DCGs in wildtype backgrounds were uniform and round, and each secretory compartment typically contained only a single large central DCG. Compartments containing abnormal and immature DCGs were therefore defined as having at least one of the following in any Z-plane apart from the first (apical) or last (basal) two in-focus Z-planes: 1) Multiple core 'fragments' present within a single compartment; 2) DCGs containing two or more acute external angles; 3) DCGs containing an internal angle greater than 180°.

The presence of Rab6-positive ILVs inside a given compartment was determined by scoring internal fluorescent puncta inside Rab6-marked compartments in *Rab6-CFP*-expressing backgrounds.

## Statistical analysis

Statistical significance for all experiments (three SCs from accessory glands from ≥10 individual males) was determined using the non-parametric Kruskal-Wallis test followed by Dunn's

multiple comparisons post hoc test with a Bonferroni correction of P values for Type 1 errors, with results from each experimental genotype being compared to results from the control. These analyses were performed on GraphPad Prism. All graphs displayed in figures show the mean value for each genotype and include error bars representing standard deviation. n ≈ 30 cells for each genotype, assessed using at least 10 independent AG lobes.

## Supporting information

**S1 Fig. Secretory compartments and ILVs in SCs are marked by several different Rab proteins (related to Fig 1).** (A) Bar chart showing the number of DCG-containing compartments in control SCs and in SCs expressing different *Rab* gene-traps, as assessed by DIC microscopy. (B-D) ILVs in SCs associate with DCGs and can form chains extending from the limiting membrane to DCGs. (B) CFP-Rab6-labelled ILVs cluster at the surface of DCGs (arrow) and form bridge-like structures which extend from the limiting membrane to the DCG boundary (arrowhead). (C) YFP-Rab11-labelled ILVs also cluster at the surface of DCGs (arrow) and form bridge-like structures which extend from the limiting membrane to the DCG boundary (arrowhead). (D) SC from *YFP-Rab11* male maintained for 12 days at 19˚C following eclosion (at 19˚C, the accessory gland matures at about half the speed of maturation at 29˚C). (E) SC from *CFP-Rab6* male maintained for 12 days at 19˚C following eclosion. (F, G) Bar charts showing number of Rab6- and Rab11-marked large compartments (F) and DCG compartments (G) in SCs from adults aged at 19˚C and 29˚C. (H) Clusters of DCG-associated ILVs (white arrow) and ILV chains (arrowhead) can be co-labelled by CFP-Rab6 and YFP-Rab11. Note that there are Rab6-positive puncta inside Rab11-compartments that are Rab6-negative (red arrow). (I) In addition to two or three DCG compartments (eg. arrow), YFP-Rab19 marks microdomains on the surface of CFP-Rab6-labelled compartments, indicated by arrowheads. Data for the bar chart were collected from three SCs per gland derived from 10 glands; bars show mean ± SD. Approximate outlines of SCs are marked by dashed circles. Scale bars: 10 μm. For A, F and G, bars show mean ± SD; CFP-Rab6, n = 34; YFP-Rab11, n = 34; YFP-Rab19, n = 30; Rab6 19C, n = 21; Rab6 29˚C, n = 30; Rab11 19˚C, n = 25; Rab11 29˚C, n = 37. Genotypes for images: (B) $w^{1118}$; $TI\{TI\}Rab6^{CFP}$/+; (C) $w^{1118}$; $TI\{TI\}Rab11^{EYFP}$/+; (D) $w^{1118}$; $P\{tub\text{-}GAL80^{ts}\}$, $TI\{TI\}Rab6^{CFP}/P\{ry^{TRiP.HMS02827}\}$; $dsx\text{-}GAL4$/+; (E) $w^{1118}$; $P\{tub\text{-}GAL80ts\}/P\{ry^{TRiP.HMS02827}\}$; $dsx\text{-}GAL4$, $TI\{TI\}Rab11^{EYFP}$/+; (H) $w^{1118}$; $TI\{TI\}Rab6^{CFP}$/+; $TI\{TI\}Rab11^{EYFP}$/+; (I) $w^{1118}$; $TI\{TI\}Rab6^{CFP}$/+; $TI\{TI\}Rab19^{EYFP}$/+. (PDF)

**S2 Fig. Rab1 and Rab2 partially colocalise with the cis-Golgi marker GM130 (related to Fig 1).** (A-F) Representative images of fixed SCs and surrounding main cells which have been immunolabelled for the cis-Golgi protein GM130 and which express either the *YFP-Rab1* (A-C) or *YFP-Rab2* (D-F) fusion proteins from the endogenous *Rab* gene locus. Panels B and E, and C and F represent magnified regions of interest (marked by boxes in A and D) from SCs and the surrounding main cells respectively. Cellular organisation was assessed through wide-field imaging of YFP fluorescence, GM130 immunolabelling, and a merged view of both. (A) Rab1 and GM130 show relatively extensive colocalization within SCs and main cells with most GM130 staining seen in a region close to the centre of the SC. (B) Within SCs, Rab1 and GM130 colocalise on punctate and tubular structures (e.g. white arrows), which presumably represent *cis*-Golgi compartments. However, Rab1 is also present in many other adjacent compartments not marked with GM130 (e.g. white arrowhead), which likely represent the medial- and *trans*-Golgi. (C) Within main cells, Rab1 and GM130 also co-localise, but in larger, more tubular structures, and there are some adjacent regions in which only Rab1 is observed. (D) Rab2 and GM130 display extensive colocalization within SCs and main cells. (E) Inside SCs,

Rab2 and GM130 share a very similar distribution, both featuring on punctate and tubular compartments near the cell centre. A limited level of adjacent Rab2-only fluorescence is also observed. (F) In main cells, Rab2 and GM130 also strongly co-localise with only a few adjacent compartments labelled by Rab2 only. Genotype for images: (A-C) $w^{1118}$; $TI\{TI\}Rab1^{EYFP}/+$; (D-F) $w^{1118}$; $TI\{TI\}Rab2^{EYFP}/+$.

(PDF)

**S3 Fig.** *Trans*-Golgi compartments co-marked by Rab1 and Rab6 can fuse during the Rab1 to Rab6 transition (related to S2 Movie). Panel shows *ex vivo* images of a single SC taken at five discrete timepoints with time since first image shown above in minutes. Rows within panel display cellular organisation at each timepoint through DIC imaging (A-E), fluorescent YFP-Rab1 signal (A'-E'), fluorescent CFP-Rab6 signal (A"-E"), and combined images display-ing all three (A"'-E"'). Two compartments marked by both YFP-Rab1 and CFP-Rab6 are marked by a white arrow and a red arrow. After the fusion of these compartments, the com-bined compartment is denoted by a red arrow with a white outline. (A-A"') The two central spherical compartments, which are jointly labelled by YFP-Rab1 and CFP-Rab6, initiated their Rab1 to Rab6 transition and expansion in volume approximately 25 minutes prior to time 0 (see S2 Movie). (B-B"') After 12 minutes, the two compartments have moved adjacent to each other, as YFP-Rab1 staining gradually diminishes. (C-C"') The two compartments fuse to form a single, larger compartment with a distorted shape. (D-D"' and E-E"') As the time-lapse video continues, the newly formed enlarged compartment regains a spherical shape and loses all YFP-Rab1 labelling. Labelling by CFP-Rab6 continues to increase, resulting in a central, spherical, Rab6-positive compartment which contains no DCG. Approximate outlines of SCs are marked by dashed circles. Scale bars: 10 μm. Genotype for images: $w^{1118}$; $TI\{TI\}Rab6^{CFP}/+$; $TI\{TI\}Rab1^{EYFP}/+$.

(PDF)

**S4 Fig. Stages of the Rab6 to Rab11 transition on SC secretory compartments (related to S4 Movie).** (A-C) Images showing progression of events in the Rab6 to Rab11 transition in SCs expressing the *YFP-Rab11* and *CFP-Rab6* gene-traps. White arrows highlight a single maturing compartment at three different timepoints during the transition. (A) Prior to Rab11 accumulation, compartments are marked by Rab6 only and are typically spherical with no ILVs present. (B) Following this, Rab11 starts to accumulate on compartment membranes at low levels. Simultaneously, compartments reduce in size and ILV biogenesis (marked by inter-nal CFP-Rab6 puncta in particular) begins. (C) Over the course of many hours, Rab6 is gradu-ally replaced by Rab11 as the primary marker of these secretory compartments; CFP-Rab6 remains visible on ILVs within compartments. Approximate outlines of SCs are marked by dashed circles. Scale bars: 10 μm. Genotype for images: $w^{1118}$; $TI\{TI\}Rab6^{CFP}/+$; $TI\{TI\}Rab11^{EYFP}/+$.

(PDF)

**S5 Fig. Expression of a second *Arf1* RNAi disrupts DCG biogenesis and Rab6/Rab11 com-partment organisation (related to Figs 4, 5 and 6).** (A-F) Representative images of SCs expressing a control RNAi or *Arf1* RNAi #2 in *GFP-GPI* (A, B), *YFP-Rab11* (C, D), and *CFP-Rab6* (E, F) backgrounds. (B) SCs expressing *Arf1* RNAi #2 contain significantly fewer DCGs than controls (A). (D) Rab11-positive compartment organisation is severely disrupted in SCs expressing *Arf1* RNAi #2 versus control (C). (F) Rab6-positive compartment organisa-tion is severely disrupted in SCs expressing *Arf1* RNAi #2 versus control. (G, H) Representative images of SCs expressing the *YFP-Rab2* gene-trap either alone (G) or alongside *Arf1* RNAi #2 (H). When compared to control SCs, the distribution of YFP-Rab2 does not appear to

significantly change following *Arf1* knockdown and YFP-Rab2 does not mark any large non-acidic compartments. Approximate outlines of SCs are marked by dashed circles. Scale bars: 10 μm. Genotypes for images: (A) $w^{1118}$; *P{tub-GAL80$^{ts}$}/P{ry$^{TRiP.HMS02827}$}; dsx-GAL4, P{UAS-GFP.GPI}/+*; (B) $w^{1118}$; *P{tub-GAL80$^{ts}$}/P{Arf1$^{KK101396}$}; dsx-GAL4, P{UAS-GFP.GPI}/+*; (C) $w^{1118}$; *P{tub-GAL80$^{ts}$}/P{ry$^{TRiP.HMS02827}$}; dsx-GAL4, TI{TI}Rab11$^{EYFP}$/+*; (D) $w^{1118}$; *P{tub-GAL80$^{ts}$}/P{Arf1$^{KK101396}$}; dsx-GAL4, TI{TI}Rab11$^{EYFP}$/+*; (E) $w^{1118}$; *P{tub-GAL80$^{ts}$}, TI{TI}Rab6$^{CFP}$/P{ry$^{TRiP.HMS02827}$}; dsx-GAL4/+*; (F) $w^{1118}$; *P{tub-GAL80$^{ts}$}, TI{TI}Rab6$^{CFP}$/P{Arf1$^{KK101396}$}; dsx-GAL4/+*; (G) $w^{1118}$; *P{tub-GAL80$^{ts}$}, TI{TI}Rab2$^{EYFP}$/P{ry$^{TRiP.HMS02827}$}; dsx-GAL4/+*; (H) $w^{1118}$; *P{tub-GAL80$^{ts}$}, TI{TI}Rab2$^{EYFP}$/P{Arf1$^{KK101396}$}; dsx-GAL4/+*. (PDF)

**S6 Fig. Knockdown of *Arf1* and *AP-1* subunits affects GFP-GPI trafficking to acidic compartments and the identity of non-acidic compartments in SCs (related to Figs 4, 5 and 6).** (A) Bar chart showing the proportion of acidic-compartments containing unquenched GFP in control SCs or following knockdown of *Arf1* or *AP-1* subunits. (B) Bar chart showing the proportion of Rab11-positive compartments which contain DCGs in these different genotypes. Note that despite the significant decrease in DCGs seen in knockdowns (Figs 4F and 5G), the proportion of Rab11-positive compartments which contained DCGs was not significantly affected by any knockdown other than *AP-1γ*, although most of these DCGs were irregularly shaped (Fig 5H). Therefore, transition to Rab11 identity typically appears to be associated with subsequent DCG formation. (C) Bar chart showing the proportion of Rab6-positive compartments which contain DCGs in these different genotypes. (D) Bar chart showing that knockdown of *Arf1* affects the Rab6-positive identity of large non-acidic compartments that do not contain DCGs. For A-D, bars show mean ± SD. For A, Control, n = 28; *AP-1γ*, n = 28; *Arf1* #1, n = 31; *Arf1* #2, n = 30; *AP-1μ*, n = 32; *AP-1σ*, n = 30. For B, Control, n = 35; *Arf1* #1, n = 31; *Arf1* #2, n = 36; *AP-1γ*, n = 30; *AP-1μ*, n = 34; *AP-1σ*, n = 30. For C, Control, n = 30; *Arf1* #1, n = 39; *Arf1* #2, n = 32; *AP-1γ*, n = 29; *AP-1μ*, n = 27; *AP-1σ*, n = 30. For D, Control, n = 30; *Arf1* #1, n = 39; *Arf1* #2, n = 32. $P<0.001$: *** $P<0.0001$: ****. (PDF)

**S7 Fig. Expression of a second *Rab6* and *Rab11* RNAi disrupts DCG biogenesis (related to Fig 7).** (A-C) Representative images of SCs expressing the DCG marker GFP-GPI with a control RNAi (A) or RNAis (#2) targeting *Rab6* (B) or *Rab11* (C). Cellular organisation is assessed through DIC imaging, GFP-GPI fluorescence and Lysotracker Red fluorescence, as well as a merged image for each cell. In both the *Rab6* and the *Rab11* knockdown, significantly fewer DCGs are present and marked by GFP-GPI. (D) Bar chart showing the proportion of acidic compartments containing unquenched GFP in these different genotypes. Approximate outlines of SCs are marked by dashed circles. Scale bars: 10 μm. For D, bars show mean ± SD, Control, n = 29; *Rab6* #1, n = 29; *Rab6* #2, n = 30; *Rab11* #1, n = 30; *Rab11* #2, n = 31. $P<0.05$: * $P<0.01$: ** $P<0.001$: *** $P<0.0001$: ****. Genotypes for images: (A) $w^{1118}$; *P{tub-GAL80ts}/P{ry$^{TRiP.HMS02827}$}; dsx-GAL4, P{UAS-GFP-GPI}/+*; (B) $w^{1118}$; *P{tub-GAL80ts}/+; dsx-GAL4, P{UAS-GFP-GPI}/P{Rab6$^{TRiP.JF02640}$}*; (C) $w^{1118}$; *P{tub-GAL80ts}/P{Rab11$^{KK108297}$}; dsx-GAL4, P{UAS-GFP-GPI}/+*. (PDF)

**S8 Fig. Expression of a dominant-negative Rab6 protein can affect DCG formation (related to Fig 7).** (A-C) Representative images of SCs expressing UAS-*YFP-Rab6* constructs encoding wild type (A), constitutively active (B) or dominant-negative (C) Rab6. (D-G) Representative images of SCs expressing UAS-*YFP-Rab6* constructs encoding wildtype (D), constitutively active (E) or dominant-negative (F, G) Rab6 in a CFP-Rab6 background. Cellular organisation

was assessed through DIC imaging, YFP-Rab6 fluorescence, CFP-Rab6 fluorescence (if present), and a merged view of all channels. (A and D) Wild type YFP-Rab6 has a similar distribution to CFP-Rab6 expressed from the endogenous *Rab6* gene locus (Fig 1C and 1D). (B and E) A constitutively active form of YFP-Rab6 shows a similar distribution pattern to wild type YFP-Rab6, but labels less secretory compartments. (C) Expressing a dominant-negative form of YFP-Rab6 produces SCs with many small compartments that do not contain DCGs. The limiting membrane of these compartments do not appear to be labelled by YFP-Rab6, some of which is abnormally sequestered into the two SC nuclei. (F, G) When the dominant-negative form of YFP-Rab6 is expressed in the CFP-Rab6 background cells, only a minority of SCs contain many small non-DCG compartments, which are not labelled by either fluorescent Rab (G). The majority of SCs in this genetic background instead produce a mix of non-DCG and DCG compartments that are mostly co-labelled with CFP-Rab6 and dominant-negative YFP-Rab6. A few compartments are labelled exclusively by one or other fluorescent marker (see the white and magenta arrows in F). (H-M) Bar charts showing the number of large secretory compartments, DCG compartments labelled by each YFP-Rab6 construct and total DCG compartments in each cell. (H-J) Compartments were counted in the absence of the endogenously labelled CFP-Rab6 protein, except for the final condition (YFP-Rab6-DN and CFP-Rab6), which leads to a highly variable phenotype. The bar charts show numbers of YFP-Rab6-labelled compartments (H), number of YFP-Rab6-labelled DCG compartments (I) and total number of DCG compartments (J). (K-M) The bar charts show numbers of YFP-Rab6-labelled compartments (K), number of YFP-Rab6-labelled DCG compartments (L) and total number of DCG compartments (M) in the absence and presence of endogenously labelled CFP-Rab6. Only the phenotype induced by YFP-Rab6-DN is affected by co-expression of CFP-Rab6. For H-M, bars show mean ± SD. For H-J, UAS-*YFP-Rab6-WT*, n = 21; UAS-*YFP-Rab6-CA*, n = 29; Driver x *UAS-YFP-Rab6-DN*, n = 17; Driver+*CFP-Rab6* x UAS-*YFP-Rab6-DN*, n = 13. For K-M, Driver x UAS-*(YFP-)Rab6-WT*, n = 10; Driver+*Rab6-CFP* x UAS-*(YFP-)Rab6-WT*, n = 13, Driver x UAS-*(YFP-)Rab6-CA*, n = 18; Driver+*Rab6-CFP* x UAS-*(YFP-)Rab6-CA*, n = 11; Driver x UAS-*(YFP-)Rab6-DN*, n = 17; Driver+*CFP-Rab6* x UAS-*(YFP-)Rab6-DN* = 13. Genotypes for images: (A) $w^{1118}$; P{tub-GAL80$^{ts}$}/P{UAS-YFP-Rab6}; dsx-GAL4/+; (B) $w^{1118}$; P{tub-GAL80$^{ts}$}/+; dsx-GAL4/P{UAS-YFP-Rab6Q71L}; (C) $w^{1118}$; P{tub-GAL80$^{ts}$}/+; dsx-GAL4/P{UAS-YFP-Rab6T26N}; (D) $w^{1118}$; P{tub-GAL80$^{ts}$}, TI{TI}Rab6$^{CFP}$/P{UAS-YFP-Rab6}; dsx-GAL4/+; (E) $w^{1118}$; P{tub-GAL80$^{ts}$}, TI{TI}Rab6$^{CFP}$/+; dsx-GAL4/P{UAS-YFP-Rab6Q71L}; (F, G) $w^{1118}$; P{tub-GAL80$^{ts}$}, TI{TI}Rab6$^{CFP}$/+; dsx-GAL4/P{UAS-YFP-Rab6T26N}.
(PDF)

**S9 Fig. Expression of wildtype, constitutively active and dominant-negative Rab11 constructs does not affect DCG formation (related to Fig 7).** (A-C) Representative images of SCs expressing UAS-*YFP-Rab11* fusion constructs that encode wild type (A), constitutively active (B) or dominant-negative (C) Rab11. (D-F) Representative images of SCs expressing UAS-*YFP-Rab11* fusion constructs that encode wild type (D), constitutively active (E) or dominant-negative (F) Rab11 in a *CFP-Rab6* background. (G) Representative image of an SC expressing the dominant-negative YFP-Rab11 construct together with the YFP-Rab11 fusion produced from the endogenous *Rab11* locus. Cellular organisation was assessed through DIC imaging, YFP fluorescence, CFP-Rab6 fluorescence if present, and a merged view of all channels. (A, D) The wild type UAS-driven YFP-Rab11 protein labels all DCG compartments and occasional non-DCG compartments when expressed in SCs, broadly matching the pattern seen with endogenously expressed YFP-Rab11 (Fig 1E). (B, E) The constitutively active YFP-Rab11 construct appears to be only weakly expressed and does not noticeably affect the

organisation of SCs. (C, F) The dominant-negative form of YFP-Rab11 does not localise to large non-acidic compartments in SCs, and is instead present at low levels throughout the cytosol and partially concentrated in faintly labelled clusters with a distribution similar to the YFP-Rab1 and YFP-Rab2 fusion proteins. The compartmental organisation of SCs is unaffected. (G) Co-expression of dominant-negative YFP-Rab11 with wild type YFP-Rab11 expressed from the endogenous *Rab11* locus reveals labelling of DCG compartments, presumably by the wild type protein. Note the subdomains of concentrated YFP-Rab11 at the outer surface of some of these compartments, which are not observed in controls. (H) Bar chart showing the number of large non-acidic compartments labelled by each YFP-Rab11 fusion protein. (I) Bar chart showing the total number of DCGs per SC following expression of each YFP-Rab11-fusion protein, as assessed by DIC microscopy. For H and I, bars show mean ± SD. UAS-*YFP-Rab11-WT*, n = 31; UAS-*YFP-Rab11-CA*, n = 9; UAS-*YFP-Rab11-DN*, n = 20. Genotypes for images: (A) $w^{1118}$; P{tub-GAL80$^{ts}$}/P{UAS-YFP-Rab11}; dsx-GAL4/+; (B) $w^{1118}$; P{UAS-YFP-Rab11Q70L}; P{tub-GAL80$^{ts}$}/+; dsx-GAL4/+; (C) $w^{1118}$; P{tub-GAL80$^{ts}$}/+; dsx-GAL4/P{UAS-YFP-Rab11S25N}; (D) $w^{1118}$; P{tub-GAL80$^{ts}$}, TI{TI}Rab6$^{CFP}$/P{UAS-YFP-Rab11}; dsx-GAL4/+; (E) $w^{1118}$; P{UAS-YFP-Rab11$^{Q70L}$}; P{tub-GAL80$^{ts}$}, TI{TI}Rab6$^{CFP}$/+; dsx-GAL4/+; (F) $w^{1118}$; P{tub-GAL80$^{ts}$}, TI{TI}Rab6$^{CFP}$/+; dsx-GAL4/P{UAS-YFP-Rab11S25N}; (G) $w^{1118}$; P{tub-GAL80$^{ts}$}/+; dsx-GAL4, TI{TI}Rab11$^{EYFP}$/P{UAS-YFP-Rab11S25N}.
(PDF)

**S10 Fig. Expression of a second *Rab6* and *Rab11* RNAi affects Rab6- and Rab11-positive compartment organisation (related to Fig 8).** (A-C) Representative images of SCs expressing the *YFP-Rab11* fusion gene from the endogenous *Rab* locus together with a control RNAi (A) or RNAis targeting *Rab6* (B) or *Rab11* (C). (D-F) Representative images of SCs expressing the *CFP-Rab6* fusion gene from the endogenous *Rab* locus together with a control RNAi (D) or RNAis targeting *Rab6* (E) or *Rab11* (F). Cellular organisation in all genotypes is assessed through DIC imaging, tagged Rab fluorescence and Lysotracker Red fluorescence, as well as a merged image for each cell. Note that some YFP-Rab11 fluorescence is still visible even after knockdown of *Rab11* (C). Approximate outlines of SCs are marked by dashed circles. Scale bars: 10 μm. Genotypes for images: (A) $w^{1118}$; P{tub-GAL80ts}/P{ry$^{TRiP.HMS02827}$}; dsx-GAL4, TI{TI}Rab11$^{EYFP}$/+; (B) $w^{1118}$; P{tub-GAL80ts}/+; dsx-GAL4, TI{TI}Rab11$^{EYFP}$/P{Rab6$^{TRiP.JF02640}$}; (C) $w^{1118}$; P{tub-GAL80ts}/P{Rab11$^{KK108297}$}; dsx-GAL4, TI{TI}Rab11$^{EYFP}$/+; (D) $w^{1118}$; P{tub-GAL80$^{ts}$}, TI{TI}Rab6$^{CFP}$/P{ry$^{TRiP.HMS02827}$}; dsx-GAL4/+; (E) $w^{1118}$; P{tub-GAL80$^{ts}$}, TI{TI}Rab6$^{CFP}$/+; dsx-GAL4/P{Rab6$^{TRiP.JF02640}$}; (F) $w^{1118}$; P{tub-GAL80$^{ts}$}, TI{TI}Rab6$^{CFP}$/P{Rab11$^{KK108297}$}; dsx-GAL4/+.
(PDF)

**S1 Movie. (related to Fig 2).** Rab1 to Rab6 transition.
(AVI)

**S2 Movie. (related to S3 Fig).** Fusion of Rab1/Rab6 co-labelled compartments.
(AVI)

**S3 Movie. (related to Fig 3).** Rab6 to Rab11 transition and DCG biogenesis.
(AVI)

**S4 Movie. (related to S4 Fig).** Rab6 to Rab11 transition.
(AVI)

**S1 Data. Datasets for Figures and Supplementary Information.**
(XLSX)

## Acknowledgments

We are extremely grateful to B. Kroeger, who initiated a significant part of the work presented. We thank all the staff at Micron Advanced Bioimaging Unit where imaging was undertaken. We also thank S. Eaton, S. Goodwin, E. Prince and F. Karch, as well as the Bloomington and Vienna *Drosophila* Stock Centres for *Drosophila* stocks. For the purpose of Open Access, the author has applied a CC BY public copyright licence to any Author Accepted Manuscript (AAM) version arising from this submission.

## Author Contributions

**Conceptualization:** Adam Wells, Cláudia C. Mendes, Felix Castellanos, Adrian L. Harris, Deborah C. I. Goberdhan, Clive Wilson.

**Data curation:** Adam Wells, Cláudia C. Mendes, Felix Castellanos, Phoebe Mountain, Tia Wright.

**Formal analysis:** Adam Wells, Cláudia C. Mendes, Felix Castellanos, Phoebe Mountain, Tia Wright, Clive Wilson.

**Funding acquisition:** Adam Wells, Adrian L. Harris, Deborah C. I. Goberdhan, Clive Wilson.

**Investigation:** Adam Wells, Cláudia C. Mendes, Felix Castellanos, Phoebe Mountain, Tia Wright, S. Mark Wainwright, M. Irina Stefana.

**Methodology:** Adam Wells, Cláudia C. Mendes, Felix Castellanos, S. Mark Wainwright, M. Irina Stefana.

**Project administration:** Deborah C. I. Goberdhan, Clive Wilson.

**Resources:** Adam Wells, Cláudia C. Mendes, S. Mark Wainwright, M. Irina Stefana.

**Supervision:** Adam Wells, S. Mark Wainwright, Adrian L. Harris, Deborah C. I. Goberdhan, Clive Wilson.

**Validation:** Adam Wells, Cláudia C. Mendes.

**Visualization:** Adam Wells, Cláudia C. Mendes, Felix Castellanos, Phoebe Mountain, Tia Wright, S. Mark Wainwright, M. Irina Stefana, Clive Wilson.

**Writing – original draft:** Adam Wells, Clive Wilson.

**Writing – review & editing:** Adam Wells, Cláudia C. Mendes, Felix Castellanos, Phoebe Mountain, Tia Wright, S. Mark Wainwright, M. Irina Stefana, Adrian L. Harris, Deborah C. I. Goberdhan, Clive Wilson.

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
