## [Decision Letter · Decision Letter 0]

1 Jun 2023

Dear Dr Wilson,

Thank you very much for submitting your Research Article entitled 'A Rab6 to Rab11 transition is required for dense-core granule and exosome biogenesis in Drosophila secondary cells' to PLOS Genetics.

The manuscript was fully evaluated at the editorial level and by independent peer reviewers. The reviewers appreciated the attention to an important problem, but raised some substantial concerns about the current manuscript. Based on the reviews, we will not be able to accept this version of the manuscript, but we would be willing to review a much-revised version. We cannot, of course, promise publication at that time.

If you decide to revise the manuscript for further consideration at PLOS Genetics, please aim to resubmit within the next 60 days, unless it will take extra time to address the concerns of the reviewers, in which case we would appreciate an expected resubmission date by email to plosgenetics@plos.org.

We are sorry that we cannot be more positive about your manuscript at this stage. Please do not hesitate to contact us if you have any concerns or questions.

Yours sincerely,

Pablo Wappner

Academic Editor

PLOS Genetics

Gregory Barsh

Editor-in-Chief

PLOS Genetics

Reviewer's Responses to Questions

**Comments to the Authors:**

Reviewer #1: Attached

Reviewer #2: In this manuscript by Wells et al, the authors examine mechanisms of biogenesis and maturation of secretory granules in the secondary cells of the Drosophila testis. The authors show that maturing secretory granules undergo sequential changes in Rab protein occupancy on the vesicle surface and suggest that canonical/conserved regulators of dense core granule biogenesis regulate this process. The images and associated quantification are high quality and convincingly show the sequence of Rab protein occupancy on the dense core vesicle membranes. However, there are several significant concerns regarding the rigor and interpretation of RNAi experiments presented in the manuscript that need to be addressed.

Major concerns:

• No validation of RNAi knockdown efficiency, either by qPCR or by western blot/imaging, is presented in the manuscript. This leads to a perceived lack of experimental rigor and calls into question the validity of the presented RNAi results.

• Figs. 5F and 6H: How do the authors explain the differing findings with the two Arf1-RNAi lines?

• “The AP-1� knockdown, in which, unlike other knockdowns, the average number of Rab6-positive compartments was not reduced compared to controls (Fig. 6H), was particularly notable. The number of Rab11-positive compartments was strongly reduced (Fig. 5F), and, in most cells, none of the compartments contained DCGs (Fig. 6I). This is consistent with AP-1 playing a key role in the Rab6 to Rab11 transition, and the hypothesis that this transition is required for DCG biogenesis.”

-- This argument by the authors does not make sense, because the three AP-1 subunits examined all form a single complex. Thus, knockdowns of all three should show similar phenotypes, unless there is a substantial difference in knockdown efficiency between the three RNAi constructs. How do the authors explain the different phenotypes observed in AP-1gamma knockdown vs. mu and sigma?

• “Our data suggest that Arf1 acts earlier in the process than AP-1, since unlike AP-1, it is required to recruit Rab6 to large compartments, as well as for the later conversion to Rab11-positive compartments. By contrast, AP-1 is primarily involved in the Rab6 to Rab11 transition and in subsequent maturation events that induce DCG formation.”

-- It is difficult to understand how the authors justify these conclusions, given that the phenotypes presented for Arf1 and for AP-1 knockdown are similar… knockdown of Arf1 and AP-1 all reduce the number of Rab11-containing compartments (Fig. 5F) and the number of Rab6-containing compartments (Fig. 6H, with the exception of AP-1gamma; see the point above). If the authors wish to make this statement, additional experiments will be necessary to justify the stated sequential roles of each protein. Alternatively, the authors could change their conclusions to accurately reflect the shown data.

Minor concerns:

• Full genotypes should be included in the figure legends, since it is unclear if the authors are imaging homozygotes or heterozygotes for tagged proteins. For example, presumably imaging of double-labeled (e.g. CFP-Rab6 with YFP-Rab11) cells is done in heterozygotes; thus, imaging of the single tags should be done in heterozygotes, as well, since 1 vs. 2 copies of the labeled proteins could affect expression levels and thus localization.

• Fig. 1G-H: How do the authors know Rab1 and Rab2 are labeling Golgi without an independent marker?

• Fig. 1K,L: What criteria were used to define a compartment as having/not having Rab6 or Rab11? The levels of these proteins appear to vary around different compartments… e.g. C’, D’, F’ all have some compartments with bright signal and others faint.

• Fig. 6G: The authors state in the text that the presence of enlarged YFP-Rab1 compartments upon knockdown of Arf1 means that these compartments failed to transition to Rab6-positive compartments. This should be directly shown using live-cell imaging.

• Reproducibility details should be included in all figure legends.

Reviewer #3: This manuscript describes the dense-core granule (DCG) formation process in the prostate-like secondary cells (SC) of the Drosophila male accessory gland. The relative ease in observing large DCGs combined with the genetic tractability of the fly provides a unique opportunity to understand the molecular events involved in DCG biogenesis in an in vivo context. The authors study the biogenesis pathway by observing the localization of different Rab GTPases and find that DCGs, marked by Rab11, originate from small precursor Rab1-labelled compartments. These compartments gradually acquire Rab6 as they mature and eventually acquire Rab11. The trafficking regulators Arf1 and AP-1 are crucial to this process, as the knockdown of these genes in secondary cells leads to failure in DCG production.

Overall, the rationale and experiments are clearly defined, and the model proposed by the authors is well supported by their experiments and is consistent with the literature.

Here is a list of minor issues that I feel should be addressed prior to publication:

1. Figures:

a. Most of the panels and graphs are labeled in a font with serifs, which makes the letters and words difficult to read. Please use a bold sans-serif font (Arial, Helvetica, etc.) throughout for clarity, unless the journal insists otherwise.

b. The scale bars throughout are very thin and in some cases barely visible. It would be helpful to thicken the bars to make them easier to see.

c. Yellow text is not visible on white when printed (Fig. 1E-J, Fig. S1C-E). Black labels could be used instead of yellow, as in Fig. 3. It can be made clear in the legends that the YFP-Rab is shown in yellow in the merged images.

d. The graphs shown in the manuscript are bar charts, not histograms. Figure legends should be corrected to reflect this.

e. The arrows, arrowheads and asterisks in some of the figures are so small that they are nearly impossible to see when printed. This is especially true in Fig. 1, but also to a lesser extent in Fig. 3 and Fig. S1. In addition, the red arrows in Fig. 2E, E’’’ are difficult to see (black might provide more contrast, and the arrows could be larger). This is also true of the red arrows in the panels in Fig. S1D (magenta might provide more contrast, and the arrows could be larger).

f. In Fig. 2, it would help to label the arrow on top, “Time (minutes)”, for clarity.

g. Panel labels are missing in Fig. 3 (A-F’’’) and Fig. S2 (A-E’’’).

2. Movies: It would be helpful to the reader if the movies could be stamped with colored labels to indicate the markers being examined. In addition, the movies should include a scale bar and time stamp.

3. Methods:

a. Fly stocks: The YFP-Rab lines are not “gene traps” (which suggests a transposable element mediated insertion event). Rather, these are engineered YFP fusions that were generated by homologous recombination at the endogenous loci. The term “gene trap(s)” should be removed from the manuscript and supplementary figure legends.

b. Fly culture and handling: Six days seems quite long to leave flies at 29°C. Male flies become sterile due to spermatogenesis defects after extended incubation at 29°C. Is it possible the cell biology of the accessory gland is also affected by this extended heat-shock regimen? Are the cells/glands morphologically similar after this treatment to cells/glands from animals not exposed to heat shock?

c. Microscopy: Details about the microscope- the objective and zoom lens combination used to achieve 1000X magnification, laser lines, excitation and emission filters, and camera- are lacking. The authors should also report the acquisition frequency for time-lapse imaging.

d. Statistics: The Kruskal-Wallis test will determine if any condition significantly differs from the control. But the authors must have performed multiple post hoc pairwise testing to identify which specific pairs of conditions are significantly different. As such, the statistics section in materials and methods should include this information. In particular, the authors should clarify whether they have performed some form of correction to the p-values when undertaking multiple pairwise comparisons. This is crucial to control the type I error rate. The differences between control and test conditions are large enough that correcting the p-values for multiple comparisons, if not done already, will likely keep the interpretation the same. However, the tests need to be performed correctly and reported accordingly.

4. References: Several references that would provide relevant background or support the authors’ interpretation of their data were not cited (PMID: 21490149, 33866198, 36239631).

5. Text: The text contains many complex sentences and is rather wordy. This could be improved by editing for brevity and flow. In addition, moving references to the ends of sentences will avoid choppiness in the text. A partial list of suggested improvements to the writing is included below.

Author summary:

p. 3, line 8: suggest replacing “which” with “that”, removing the comma

p. 3, line 15: move comma in “upo,n” to say “upon,”

Introduction:

p. 4, line 3: suggest replacing “secreting” with “producing”

p. 4, line 5: add comma after “neurons”

p. 4, line 6: replace “into” with “in”

p. 6, line 1: cite PMID: 21490149, 33380435

p. 6, line 13: add citation to PMID: 34342349

Results:

p. 8, line 8: delete “in SCs”

p. 10, line 11: suggest replacing “which” with “that”

p. 12, line 6: delete comma after “compartments”

p. 12, line 23: delete “presumably” and change “persists,” to “persist”, removing the comma

p. 15, line 6: change to “Fig. 4B, C, F”

p. 18: line 10: replace “did form” with “formed”

p. 18, line 12: delete “at all. We found that” and start new sentence with “Knockdown”

p. 18, line 13: delete “which lack” and replace with “lacking”

p. 18, line 18: add comm after “knockdowns” and replace “and indicate” with “indicating”

p. 18, line 22: replace “In order to” with “To”

p. 19, line 3: delete “at all in SCs”

p. 19, line 15: replace “where” with “which”

Discussion:

p. 20, line 2: replace “The genetic” with “Genetic”

p. 20, line 15: replace “by” with “at”

p. 21, lines 2-5: suggest moving this sentence earlier in the paragraph to end on a stronger note

p. 22, line 11: delete “knockdown”

p. 22, line15: replace “which” with “that” and remove commas after “materials” and “formation”

Rab19 should be mentioned somewhere in the Introduction, as it comes as a surprise here and in the model figure.

p. 25, line 1: consider replacing “biogenesis” with “stabilization”

p. 25, line 9: Dar et al., 2021 (PMID: 34795295 is missing from the reference list

Materials and Methods:

p. 26, line 19: replace “which contained” with “containing”

p. 27, line 1: replace “could vary” with “varied”

p. 28, line 3: delete “which”

p. 28, line 10: change “uniform, round and” to “uniform and round, and”

p. 28, line 12: replace “which “contained” with “containing”

p. 28, line 19: add comma to end of sentence

p. 28, line 23: delete “the” after “representing”

6. Figure Legends: The Figure Legends would benefit from editing to remove sentences that restate the Results.

p. 39, line 24: define “large compartments” (specify greater than what diameter)

p. 41, line 21: change “compartments” to “compartment” (only one is shown)

p. 41, line 22: change “have to “has”

p. 42, line 10: specify whether the “highlighted compartments” are marked by the blue and yellow arrows

p. 47, line 5: add comma after “Rab19”

p. 47, line 12: add comma after “identity” and delete “and”

p. 47, line 13: change “which” to “and” and “and no” to “or”

p. 47, line 15: delete “their”

p. 47, line16: add comma after “membrane” and replace “which induces the” with “inducing”

p. 47, lines 21-22: delete “Redhai et al. (2016) previously demonstrated that” and start sentence with “BMP signalling”

p. 47, line 24: insert “(Redhai et al., 2016)” after “event”; also change “We additionally show here that AP-1…” to “Additionally, AP-1…”

p. 48, lines 1-2: replace “matured” with “mature” (two instances”

p. 48, line 3: replace “was previously shown” with “is”

p. 48, line 4: replace “what” with “the”, delete “are”, and change “remains” to “remain”

p. 48, line 5: delete “to be” and “compartment”

Fig. S1 legend, lines 5, 9: replace “chains” with “bridge-like structures” for consistency with the Results section

**Have all data underlying the figures and results presented in the manuscript been provided?**

Reviewer #1: Yes

Reviewer #2: Yes

Reviewer #3: Yes

PLOS authors have the option to publish the peer review history of their article (what does this mean?). If published, this will include your full peer review and any attached files.

Reviewer #1: No

Reviewer #2: No

Reviewer #3: No

---

## [Decision Letter · Decision Letter 1]

17 Sep 2023

Dear Dr Wilson,

We are pleased to inform you that your manuscript entitled "A Rab6 to Rab11 transition is required for dense-core granule and exosome biogenesis in Drosophila secondary cells" has been editorially accepted for publication in PLOS Genetics. Congratulations!

Yours sincerely,

Pablo Wappner

Academic Editor

PLOS Genetics

Gregory Barsh

Editor-in-Chief

PLOS Genetics

Comments from the reviewers (if applicable):

Reviewer's Responses to Questions

**Comments to the Authors:**

Reviewer #1: This reviewer is completely satisfied with the very detailed answers provided by the authors to each and every question raised by the three reviewers of the manuscript. In my opinion, the new version of the manuscript is clearer and the novelty of the findings much better communicated.

I appreciate the commitment with which the authors have addressed each of the reviewer´s comments, concerns and suggestions. I have no further concerns to raise.

Reviewer #2: The authors have sufficiently addressed all concerns, making the paper suitable for publication

**Have all data underlying the figures and results presented in the manuscript been provided?**

Reviewer #1: Yes

Reviewer #2: Yes

PLOS authors have the option to publish the peer review history of their article (what does this mean?). If published, this will include your full peer review and any attached files.

Reviewer #1: No

Reviewer #2: No

**Data Deposition**

http://datadryad.org/submit?journalID=pgenetics&manu=PGENETICS-D-23-00462R1

**Press Queries**

---

## [Editor Report · Acceptance letter]

11 Oct 2023

PGENETICS-D-23-00462R1 

A Rab6 to Rab11 transition is required for dense-core granule and exosome biogenesis in Drosophila secondary cells 

Dear Dr Wilson, 

We are pleased to inform you that your manuscript entitled "A Rab6 to Rab11 transition is required for dense-core granule and exosome biogenesis in Drosophila secondary cells" has been formally accepted for publication in PLOS Genetics! Your manuscript is now with our production department and you will be notified of the publication date in due course.

With kind regards,

Timea Kemeri-Szekernyes

PLOS Genetics

On behalf of:
